# Learning Coherent Representations: A Topological Approach to Interpretability

**Sigurd Gaukstad** [1]   **Melvin Vaupel** [1]   **Valdemar Kargård Olsen** [2]   **Erik Hermansen** [1]   **Benjamin A. Dunn** [1]

## Abstract

Deep neural networks learn representations where individual features often lack interpretable meaning; a single neuron may activate for scattered, unrelated inputs. We introduce coherence, a geometric property inspired by neural coding in the brain, where neurons like grid cells and head direction cells respond to contiguous regions of state space. A non-negative matrix is coherent if each row (sample) attends to geometrically clustered columns (features) and vice versa, and in addition every sample is well described by some feature and every feature is needed by some sample. We prove that coherent matrices induce a bounded interleaving between the Vietoris-Rips filtrations of samples and features, guaranteeing that both spaces share compatible topological structure. This geometric constraint facilitates interpretability. For example, if data lies on a circle, coherent features must tile that circle into contiguous arcs. We introduce COH, a differentiable objective function based on Fréchet variance that enforces coherence during training. Unlike sparsity, which bounds how many samples a feature activates on, coherence bounds *which* samples, requiring geometric connectivity rather than only rarity. This yields not just interpretable features but an interpretable feature space. We validate COH in an auto-encoder using synthetic and rotated MNIST datasets and in a token embedding of BERT using language data.

## 1. Introduction

Deep neural networks (DNNs) trained on classification tasks progressively transform data representations such that class manifolds become geometrically separable (Cohen et al., 2020). This property, that samples from the same class cluster together in latent space, emerges naturally from the classification objective and underlies the success of transfer learning and feature visualization. However, this says nothing about the features themselves: individual latent dimensions may respond to scattered, incoherent subsets of the data (Elhage et al., 2022), limiting interpretability. For unsupervised approaches such as auto-encoders, the situation is even worse. Without class labels to guide separation, networks can learn representations where semantically related samples are spread across the latent space, and where individual features lack any coherent meaning. Sparsity regularization ($L^1$) encourages features to activate on few samples, but does not ensure those samples are geometrically related, a sparse feature may fire on disconnected regions of the data manifold. Strikingly, biological neural circuits achieve interpretable representations without explicit supervision. Grid cells in the entorhinal cortex tile physical space with periodic firing fields (Hafting et al., 2005; Gardner et al., 2022). Head direction cells activate for specific orientations, with each neuron covering a contiguous arc of angles (Taube et al., 1990; Rybakken et al., 2019). These neural codes exhibit locality: each neuron's activity is concentrated on a coherent region of the underlying state space. This locality is precisely what makes these cells interpretable, one can read off an animal's position or heading from the concurrent neural activity because the mapping between state and response is geometrically organized. This observation suggests that locality is not merely a byproduct of evolutionary optimization, but a design principle for interpretable neural codes. Can we formalize this principle and impose it on artificial neural networks?

We focus primarily on unsupervised settings — autoencoder bottlenecks and transformer token embeddings — where coherence regularization must induce structure without label guidance. In supervised classifiers, cross-entropy loss naturally collapses within-class variation, leaving little structure for coherence to preserve such that the resulting topology is approximately discrete. Thus, applying the method that we develop in this work to supervised settings where within-class structure matters is a direction for future work. Aside from the observations of neural activity in the brain, this work is greatly inspired by the Dowker Duality (Dowker, 1952), a famous duality between the topology of rows and

---

[1]Department of Mathematical Sciences, NTNU, N-7491 Trondheim, Norway [2]Kavli Institute for Systems Neuroscience, NTNU, Trondheim, Norway. Correspondence to: Sigurd Gaukstad <sigurd.gaukstad@ffi.no>.

*Proceedings of the 43rd International Conference on Machine Learning*, Seoul, South Korea. PMLR 306, 2026. Copyright 2026 by the author(s).

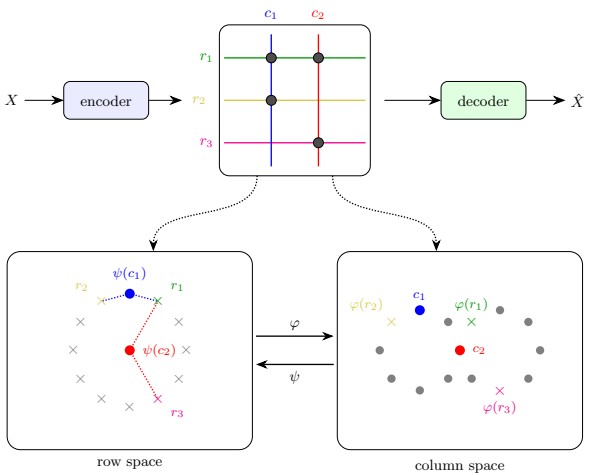

*Figure 1.* Given an auto-encoder with a non-negative activation function, we can treat the encoded latent space as a matrix $M$, whose rows are samples and columns are features. Most often, the *topology* of the samples and features are vastly different. We regularize these spaces to be topologically similar by creating an explicit interleaving between the filtered simplicial complexes induced by the latent samples and latent features by using the barycentric maps $\phi$ and $\psi$. Our definition of a matrix to be $\epsilon$-*local* implies that any sample $r_i$ is mapped $\epsilon^{1/2}$-close to some feature $c_k$. For instance, the position of their barycentric images in the weight space of all rows shows that the column $c_1$ is more local than the column $c_2$. Moreover, our definition of $\epsilon$-covering implies that around any sample $r_k$ there is some feature that maps $\epsilon^{1/2}$-close to it under the barycentric map. When a matrix $M$ (latent space) has both these properties we call it $\epsilon$-coherent, and together with a non-expanding assumption on the barycentric maps, this implies that the latent samples and latent features are topologically $\epsilon^{1/2}$-similar.

columns of a binary matrix, and we consider our work as a geometric analogue, where we do not get Dowker Duality for free, but rather have to define a class of matrices where the geometric Vietoris-Rips row and column filtrations, as defined in Section C, are similarly interleaved.

## 1.1. Contributions

1. **Definition of coherence.** We define a non-negative matrix to be $\epsilon$-*coherent* if it is both $\epsilon$-*local* (each row attends to a geometrically clustered set of columns, and vice versa) and an $\epsilon$-*covering* (each row is matched by some column's barycenter, and vice versa). We prove that coherent matrices induce a bounded interleaving between the Vietoris–Rips filtrations of samples and features, so the two spaces share compatible topology.

2. **Differentiable objective function.** We derive COH, a differentiable loss based on Fréchet variance, addable to any architecture with non-negative activations. Its

two terms penalize the locality and covering quantities from the definition above, driving representations toward $\epsilon$-coherence.

3. **Empirical validation.** We show that COH produces interpretable feature spaces across distinct settings: it recovers the expected topology in synthetic and rotated-MNIST autoencoders, and in a BERT token embedding it yields features aligned with human-readable categories (e.g., years, kinship terms, and place names, but also units of measurement, hedging adverbs, and directional prepositions), whereas non-negativity alone yields essentially none.

Our work connects topological data analysis, neuroscience, and representation learning, providing both theoretical foundations and a practical objective function for learning interpretable latent spaces.

## 1.2. Related Work

### Interpretability.

As models scale, understanding learned representations becomes critical for safety and debugging (Olah et al., 2020). A representation is interpretable if individual features correspond to human-understandable concepts.

Cohen et al. (Cohen et al., 2020) showed that classification training progressively untangles class manifolds, making them linearly separable in later layers; Mamou et al. (Mamou et al., 2020) observed similar separation in language models. These results characterize *sample* geometry but do not address whether individual features are interpretable. Sparse coding (Olshausen & Field, 1996) and recent sparse auto-encoders for mechanistic interpretability (Bricken et al., 2023; Cunningham et al., 2024) address this by encouraging features to activate rarely, reducing polysemanticity. However, sparsity constrains *how many* samples a feature activates on, not *which*—a sparse feature may fire on geometrically scattered inputs. Coherence explicitly requires that active samples be spatially clustered, providing a geometric guarantee that features align with data structure.

**Geometric deep learning.** Geometric deep learning (Bronstein et al., 2021) incorporates known symmetries (translation, rotation, permutation) into network architectures, reducing the hypothesis space and improving generalization. Our approach is complementary: rather than encoding symmetries architecturally, we regularize the learned representation to exhibit geometric structure, specifically requiring that feature and sample spaces share compatible topology.

**Topological auto-encoders.** Topological data analysis (TDA) provides robust tools for characterizing data shape (Carlsson, 2009), with stability results ensuring that

small perturbations in data yield small changes in topological descriptors (Chazal et al., 2009). Several lines of work use tools from TDA to regularize neural networks such as (Moor et al., 2020; Hofer et al., 2019; Hu et al., 2019). These approaches aim to *preserve* input topology in the latent representation or to give the output space a target topology. Our goal differs fundamentally: we do not preserve topology but *mirror* it between latent samples and latent features. Another difference is that we operate at the level of simplicial filtrations, via explicit interleaving maps, avoiding the need to choose a homology degree, which these methods are bound by.

**Neural coding.** Our work draws inspiration from neuroscience, where grid cells (Hafting et al., 2005) and head direction cells (Taube et al., 1990) exhibit local receptive fields. These local fields allow us to treat every row or column of the data matrix as a "point", revealing the topological space formed by the data, as shown for grid cells (Gardner et al., 2022) and head direction cells (Rybakken et al., 2019).

**Similarity-preserving networks.** Sengupta et al. (Sengupta et al., 2018) showed that non-negative similarity-preserving objectives yield localized receptive fields tiling input manifolds. Our work differs in requiring bidirectional coherence. The feature space must share the sample space's topology. This provides explicit interleaving guarantees rather than characterizing receptive field shapes.

## 2. Background

We briefly establish notation and point to Appendix C for full definitions. Given a finite set $P$ in a metric space, the *Vietoris-Rips filtration* $\{VR_t(P)\}_{t \geq 0}$ is a nested family of simplicial complexes capturing the topology of $P$ at increasing scales. Two filtrations are $\delta$-*interleaved* if there exist maps between them that approximately commute with the inclusions, up to a $\delta$-shift in scale (see Definition C.6). The *interleaving distance* bounds the bottleneck distance between persistence diagrams, ensuring similar homological features.

Intuitively, two spaces that are $\delta$-interleaved are 'topologically $\delta$-similar', i.e., they have the same large-scale shape and differ only at scales below $\delta$.

## 3. Coherent Matrices

The goal of this section is to introduce the concept of coherent matrices and show how it induces a natural interleaving between metric spaces associated to the rows and columns of the matrix $M$. The pairwise distances between row and column vectors respectively can lie on vastly different scales for non-square matrices. In comparing their induced topolo-

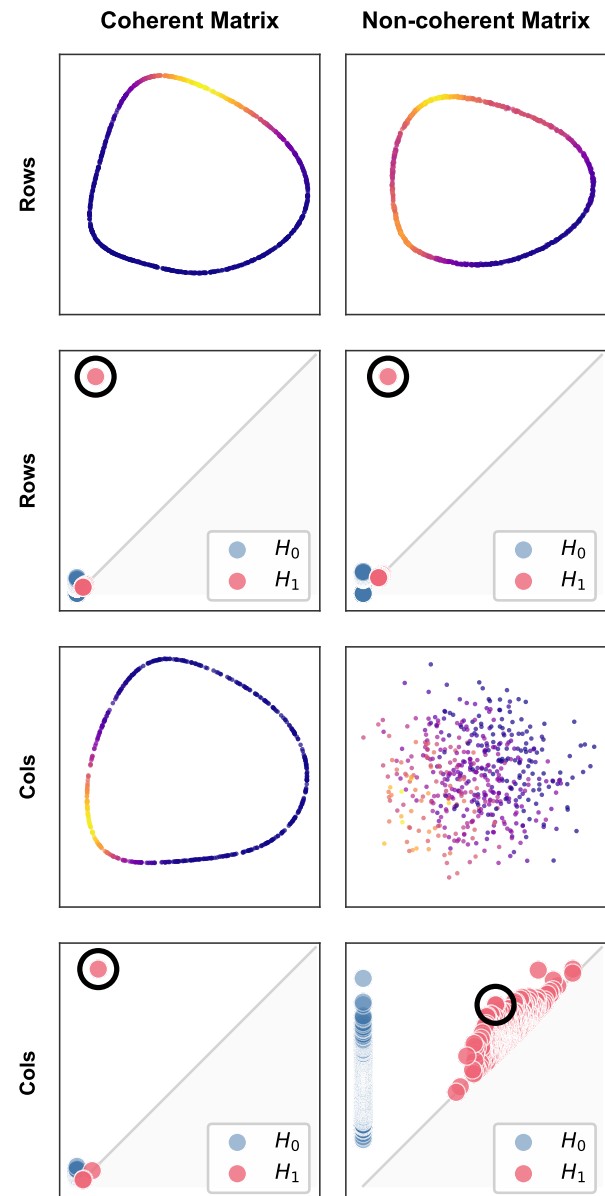

*Figure 2.* Coherent vs non-coherent matrices derived from circular state space. *Left*: Coherence $\varepsilon = 0.18$. *Right*: Non-coherent $\varepsilon = 1.46$. In rows one and three, we show PCA projection of rows and columns colored by the activation of a column and row. In row two and four we show persistence diagrams of rows and columns where we highlight the most persistent $H_1$. Note that only the coherent matrix exhibits matching circular topology in both rows and columns.

gies we want to be agnostic to this difference. For practical purposes we remedy it by normalizing the row and column metrics by scaling by a factor of one over the mean of pairwise distances. Many of the following results can be done for any $L^p$-norm, but we will stick to the Euclidean norm as this makes barycentric maps linear with a simple, closed well-defined form that is easy to compute in training. We note that the choice of the Euclidean norm may suffer in high dimensions, but is a natural choice for our method.

Throughout this section, we assume we have a non-negative matrix $M \in \mathbb{R}_+^{m \times n}$ with **no zero rows or zero columns**. We will denote by

$$\mathcal{R} = \{r_1, \ldots, r_m\} \subset \mathbb{R}^n \quad \text{and} \quad \mathcal{C} = \{c_1, \ldots, c_n\} \subset \mathbb{R}^m$$

the set of rows and columns of $M$. We want the non-negative rows and columns in the matrix to act like probability distributions over the columns and rows respectively, hence we define the notion of normalization kernels.

**Definition 3.1.** Let $\sigma_R : \mathbb{R}_+^n \setminus \{0\} \to \Delta^{n-1}$ and $\sigma_C : \mathbb{R}^m \setminus \{0\} \to \Delta^{m-1}$ be continuous functions that map a non-zero vector to a probability vector. We call $\sigma_R$ and $\sigma_C$ *normalization kernels*. Let $M \in \mathbb{R}_+^{m \times n}$. We define the *generalized weight spaces* as follows:

1. The *row-weight space* $\mathcal{W}$ is defined by the matrix $W$, where the $i$-th row is: $W_{i,\cdot} = w^{(i)} = \sigma_R(r_i)$.

2. The *column-weight space* $\mathcal{V}$ is defined by the matrix $V$, where the $j$-th row is: $V_{j,\cdot} = v^{(j)} = \sigma_C(c_j)$.

**Example 3.2.** *Examples of normalization kernels are $L^1$ normalization, softmax and squared $L^1$ normalization, which we use in our experiments.*

We now have a natural choice of maps between the row-weight space and column-weight space using the *Fréchet mean*.

**Definition 3.3.** Let $\mathcal{A} \subset \mathbb{R}^n$ be a set of vectors. We denote by $\mathrm{conv}(\mathcal{A})$, the *convex hull* of the vectors $\mathcal{A}$.

**Definition 3.4.** Let $M \in \mathbb{R}_+^{m \times n}$ and let $\{w^{(i)}\}_{i \in \{1,\ldots,m\}}$ and $\{v^{(j)}\}_{j \in \{1,\ldots,n\}}$ be row and column weights induced by some normalization kernels $\sigma_R$ and $\sigma_C$. We have the barycenters

$$\tilde{\phi} : \mathcal{R} \to \mathrm{conv}(\mathcal{C}) \quad \text{defined by} \tag{1}$$

$$r_i \mapsto \operatorname*{argmin}_{\mu \in \mathrm{conv}(\mathcal{C})} \sum_{j=1}^n w_j^{(i)} \|\mu - c_j\|_2^2 \tag{2}$$

and

$$\tilde{\psi} : \mathcal{C} \to \mathrm{conv}(\mathcal{R}) \quad \text{defined by} \tag{3}$$

$$c_j \mapsto \operatorname*{argmin}_{\mu \in \mathrm{conv}(\mathcal{R})} \sum_{i=1}^m v_i^{(j)} \|\mu - r_i\|_2^2. \tag{4}$$

We extend these maps linearly to $\phi : \mathrm{conv}(\mathcal{R}) \to \mathrm{conv}(\mathcal{C})$ and $\psi : \mathrm{conv}(\mathcal{C}) \to \mathrm{conv}(\mathcal{R})$.

*Remark* 3.5. We note that the barycenters in Definition 3.4 are well-defined, linear and have the closed form

$$\phi(r_i) = w^{(i)} M^T \quad \text{and} \quad \psi(c_j) = v^{(j)} M.$$

The barycentric maps give a natural way of going between the convex hull of the rows and the convex hull of the columns. The next definition, *Fréchet variance*, will capture how stable these maps are. This allows us to define the notion of an $\epsilon$-local matrix, where $\epsilon$ is an upper bound to the variance for all rows and columns:

**Definition 3.6.** Let $M \in \mathbb{R}_+^{m \times n}$. We define the *Fréchet variance* of a row $r_i$ of $M$ as

$$\mathrm{Var}_{\mathcal{R}}(r_i) := \sum_{j=1}^n w_j^{(i)} \|\phi(r_i) - c_j\|_2^2,$$

and the variance of a column $c_j$ of $M$ as

$$\mathrm{Var}_{\mathcal{C}}(c_j) := \sum_{i=1}^m v_i^{(j)} \|\psi(c_j) - r_i\|_2^2.$$

We say that $M$ is $\epsilon$-*local* if for any row index $i$ and column index $j$ we have that

$$\mathrm{Var}_{\mathcal{R}}(r_i) \leq \epsilon \quad \text{and} \quad \mathrm{Var}_{\mathcal{C}}(c_j) \leq \epsilon.$$

The next thing we can ask is that, given a row (or column), does there exist a column (or row) that maps close to it under the barycentric map? This corresponds to asking whether, for a given sample, does there exist a feature that describes it well? Or dually, given a feature, does there exist a sample that needs it?

**Definition 3.7.** Let $M \in \mathbb{R}_+^{m \times n}$. We define the *covering* of a row $r_i$ of $M$ as

$$\mathrm{Cov}_{\mathcal{R}}(r_i) := \sum_{j=1}^n w_j^{(i)} \|r_i - \psi(c_j)\|_2^2,$$

and the covering of a column $c_j$ of $M$ as

$$\mathrm{Cov}_{\mathcal{C}}(c_j) := \sum_{i=1}^m v_i^{(j)} \|c_j - \phi(r_i)\|_2^2.$$

We say that $M$ is $\epsilon$-*covered* if for any row index $i$ and column index $j$ we have that

$$\mathrm{Cov}_{\mathcal{R}}(r_i) \leq \epsilon \quad \text{and} \quad \mathrm{Cov}_{\mathcal{C}}(c_j) \leq \epsilon.$$

We can now show that coherence implies a bound for the interleaving distance between the rows and columns of $M$. To have an interleaving, we need a matching between the set of rows and columns, as our barycentric maps map into the convex hull, not onto a particular row or column, we have to define a hard barycentric map, by snapping to the closest row or column.

**Definition 3.8.** Let $M \in \mathbb{R}_+^{m \times n}$. We define the *snapping barycentric maps* as $\Phi : \mathcal{R} \to \mathcal{C}$ by

$$r_i \mapsto \underset{c_j \in \mathcal{C}}{\operatorname{argmin}} \|\phi(r_i) - c_j\|_2$$

and $\Psi : \mathcal{C} \to \mathcal{R}$

$$c_j \mapsto \underset{r_i \in \mathcal{R}}{\operatorname{argmin}} \|\psi(c_j) - r_i\|_2.$$

We note that there might be several closest columns and rows, in that case one makes an arbitrary choice.

Locality of a matrix $M$ gives a bound on how much the barycentric map and the barycentric snapping map can disagree. In practice this allows us to work with the differentiable barycentric maps, as we know how much is lost when we pass to the non-differentiable snapping maps that we use to make the matching.

**Proposition 3.9.** *Let* $M \in \mathbb{R}_+^{m \times n}$. *If* $M$ *is* $\epsilon$-*local, then*

$$\|\phi(r_i) - \Phi(r_i)\|_2 \le \epsilon^{1/2} \quad and \quad \|\psi(c_j) - \Psi(c_j)\|_2 \le \epsilon^{1/2}.$$

*Proof sketch.* The variance bound implies that the weighted columns $c_j$ concentrate around $\phi(r_i)$. Since $\Phi(r_i)$ is one of these columns, it lies within the variance ball. Full proof in Appendix D.

**Proposition 3.10.** *If* $M$ *is* $\epsilon$-*covered, then for any row* $r_i$ *and column* $c_j$ *the roundabout trips are bounded by* $\epsilon^{1/2}$:

$$\|r_i - \psi \circ \phi(r_i)\|_2 \le \epsilon^{1/2} \quad and \quad \|c_j - \phi \circ \psi(c_j)\|_2 \le \epsilon^{1/2}.$$

*Proof sketch.* By linearity, $\psi \circ \phi(r_i)$ is a weighted average of the barycenters $\psi(c_j)$. The covering bound ensures $r_i$ is close to each $\psi(c_j)$ in a weighted sense. Together with Jensen's inequality this implies that $r_i$ lies within $\epsilon^{1/2}$ of their weighted average. Full proof in Appendix D.

**Definition 3.11.** Let $M \in \mathbb{R}_+^{m \times n}$. We say $M$ is $\epsilon$-*coherent* if $M$ is $\epsilon$-local and $\epsilon$-covered.

See Figure 2 for an illustration of a coherent matrix versus a non-coherent matrix.

**Theorem 3.12.** *If* $M \in \mathbb{R}_+^{m \times n}$ *is* $\epsilon$-*coherent and the barycentric maps* $\phi$ *and* $\psi$ *are* 1-*Lipschitz, then the barycentric snapping maps* $\Phi$ *and* $\Psi$ *induce an* $\epsilon^{1/2}$-*interleaving between* $VR(\mathcal{R}, d_2)$ *and* $VR(\mathcal{C}, d_2)$.

*Proof sketch.* We verify the four conditions in Lemma C.9. We want to use the maps $\psi$ and $\phi$ between the rows and columns, but these land in the convex hulls rather than on actual rows or columns. Therefore, we must use the hard snapping maps $\Psi$ and $\Phi$ and pay the cost of $\epsilon^{1/2}$ given by Proposition 3.9 and Proposition 3.10.

*Remark* 3.13. The result generalizes immediately to $K$-Lipschitz barycentric maps, yielding a $K\epsilon^{1/2}$-interleaving. We state the $K = 1$ case for clarity since, in our experiments, these maps satisfy this assumption up to negligible violations.

# 4. Algorithm

See Appendix A for more details of implementation.

---

**Algorithm 1** COH

---

**Input:** Non-negative matrix $M \in \mathbb{R}_{\ge 0}^{B \times L}$ (Batch $\times$ Latent), top-$k$ parameters $k_R, k_C$, target variance $\tau$
**Output:** Coherence loss $\mathcal{L}_{\text{COH}}$

*// Compute scale factors (sampled for efficiency)*
$\bar{d}_R \leftarrow \text{MeanPairwiseDist}(\{r_i\})$
$\bar{d}_C \leftarrow \text{MeanPairwiseDist}(\{c_j\})$

*// Compute weights (Squared L1 normalization)*
$W_{ij} \leftarrow M_{ij}^2 / \sum_k M_{ik}^2$    *// Row weights*
$V_{ji} \leftarrow M_{ij}^2 / \sum_k M_{kj}^2$    *// Column weights*

*// Compute barycenters*
$\phi(r_i) \leftarrow W_{i,:} M^T$ for all $i$
$\psi(c_j) \leftarrow V_{j,:} M$ for all $j$

*// Row variance and covering (normalized)*
**for** $i = 1$ **to** $B$ **do**
    $\text{Var}_R(r_i) \leftarrow \sum_j W_{ij} \|c_j - \phi(r_i)\|^2 / \bar{d}_C^2$
    $\text{Cov}_R(r_i) \leftarrow \sum_j W_{ij} \|r_i - \psi(c_j)\|^2 / \bar{d}_R^2$
**end for**

*// Column variance and covered (normalized)*
**for** $j = 1$ **to** $L$ **do**
    $\text{Var}_C(c_j) \leftarrow \left( \sum_i V_{ji} \|r_i - \psi(c_j)\|^2 \right) / \bar{d}_R^2$
    $\text{Cov}_C(c_j) \leftarrow \left( \sum_i V_{ji} \|c_j - \phi(r_i)\|^2 \right) / \bar{d}_C^2$
**end for**

*// Top-k aggregation with threshold*
$\mathcal{L}_{\text{var}} \leftarrow \text{TopK}([\text{Var}_R - \tau]_+, k_R) + \text{TopK}([\text{Var}_C - \tau]_+, k_C)$
$\mathcal{L}_{\text{cov}} \leftarrow \text{TopK}([\text{Cov}_R - \tau]_+, k_R) + \text{TopK}([\text{Cov}_C - \tau]_+, k_C)$

**return** $\mathcal{L}_{\text{var}} + \mathcal{L}_{\text{cov}}$

---

# 5. Auto-Encoder Experiments

We evaluate COH on synthetic and real datasets with known topological structure, measuring whether learned features align with interpretable data attributes.

We compare with the plain auto-encoder without objective function and with $L^1$-regularization as the most canonical and simple way of getting interpretability. We use Ripser (Bauer, 2021) for persistent homology and UMAP (McInnes et al., 2018) for visualization.

**Datasets** For the toy data set we sample 20,000 points from the disjoint union of two circles embedded in $\mathbb{R}^{512}$. We train on half of the samples and create latent spaces from the other half. We use rotated MNIST, sampling each digit at 72 uniformly spaced angles with 250 samples per angle. We use 90% of the data for training and do all analysis on the latent space given by the latent test samples. In the single digit experiment we are looking at the digit 6 as it should be easy to have a circular latent space. For the two digit experiments, we pick 3 and 7 as those are dissimilar and without non-trivial symmetries under rotation.

**Hyperparameters.** We select $\lambda_{\text{COH}} = 10^{-3}$ and $\lambda_{L^1} = 10^{-3}$ for both MNIST experiments, balancing reconstruction loss with regularization strength. For the toy example, we set $\lambda_{\text{COH}} = 10^{-5}$ and $\lambda_{L^1} = 10^{-4}$ without systematic tuning. See Table 4 for parameter sweeps.

## 5.1. Metrics for Interpretability

To evaluate whether learned features are interpretable, we measure how well each feature's activation aligns with known structure in the data. We introduce three metrics that yield directly human-readable feature descriptions in experiments.

**Component score.** For data with $K$ discrete components/labels, we measure whether features concentrate on a single component/label:

$$\text{compscore}(c) := \frac{K}{K-1} \left( \frac{\max_k \sum_{i \in \text{component } k} c_i}{\sum_i c_i} - \frac{1}{K} \right).$$

Scores range from 0 (uniform) to 1 (within one single component). We report the fraction of features with compscore > 0.5. (Pure)

**MRL.** When data has circular structure (e.g., rotations), an interpretable feature should activate on a coherent arc. We use the Mean Resultant Length (MRL):

$$\text{MRL}(c) := \frac{\left\| \sum_i c_i e^{i\theta_i} \right\|}{\sum_i c_i},$$

where $\theta_i$ is the angle associated with sample $i$. A score of 1 indicates all activation at a single angle; a score near 0 indicates activation spread uniformly around the circle. We report the fraction of features with MRL > 0.5, indicating meaningful angular selectivity. (Tuned)

**MRL$_{180}$.** Digits 3, 6 and 7 lack 180 degree symmetry, yet networks often encode antipodal angles together. We compute MRL with doubled angles:

$$\text{MRL}_{180}(c) := \frac{\left\| \sum_i c_i e^{2i\theta_i} \right\|}{\sum_i c_i},$$

so features firing at $\theta$ and $\theta + \pi$ receive high scores rather than canceling. We report the fraction of features with MRL$_{180}$ > 0.5, indicating meaningful angular selectivity. (Tuned$_{180}$)

In addition to these metrics we keep track of how sparse the features are. We say a feature is considered active if its activation exceeds 1% of the maximum activation. We record the mean percentage of the active feature per sample.

**Toy experiment** The learned barycentric maps in the COH model satisfy the 1-Lipschitz assumption (checked by sampling $1000 \times 256$ pairs), yielding a $(0.14)^{1/2}$-interleaving by Theorem 3.12. See Figure 3 for a visualization of the resulting latent spaces and Table 1 for the results from the experiment. All three models encode a seemingly similar latent sample space. When looking at the latent features, the story is different, and it is not clear how the feature space relates to the original data except for our COH model. See Figure 5 and Figure 6 for similar results on other spaces.

**Single digit experiment** With a single rotated class (6) of digits we observe in Table 2 that COH has all features being tuned to the angles of the data, while being much less sparse than the features for $L^1$. There also is very little cost to this improvement in terms of MSE. In Figure 4 we see the duality of the circular space and in Figure 9 we observe that features and samples define activation on each others space. The average samples plotted over a feature clearly shows us the rotated expected digits. In Figure 10 we decode the persistent $H_1$'s against the true angles confirming that the found circles are given by the angles generating the data. We justify the 1-Lipschitz assumption by sampling $1000 \times 256$ pairs for $\phi$ and $\psi$ across five seeds on the COH latent space, here we find that $\psi$ satisfies the condition nearly exactly (violation rate < 0.002%) and $\phi$ violates it on $4.2 \pm 0.9\%$ of pairs with mean expansion 1.045, hence the interleaving is almost a $0.15^{1/2}$-interleaving. The high values for MRL$_{180}$ in the COH latent space are due to wide fields.

**Double digits experiment** This setting is more challenging as the latent space must represent two classes $\times 72$ angles. Our model shows high variance across seeds (Table 5)

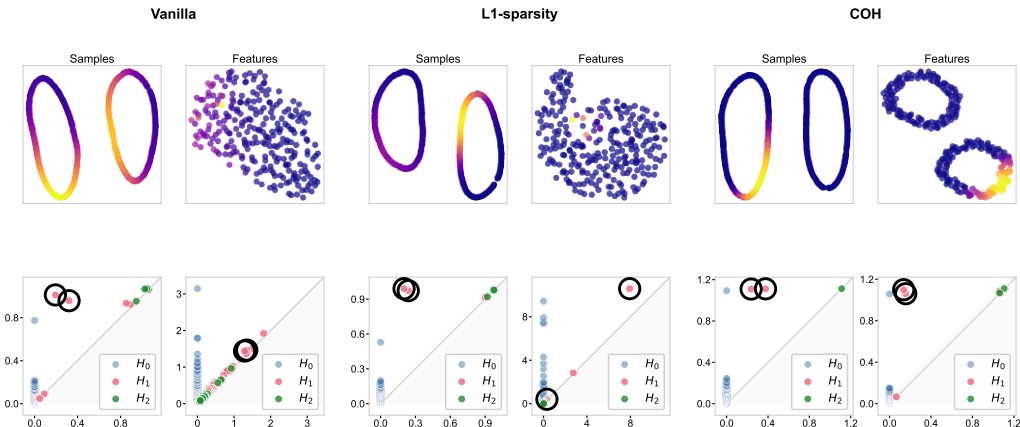

*Figure 3.* **Two Circles Toy experiment.** UMAP projections of latent samples and latent features , with persistence diagrams for each. We highlight the two most persistent $H_1$ features, representing the two circles. Samples are colored by activation of a single feature; features are colored by activation on a single sample. Only the COH model gives the expected disjoint circular topology in both spaces.

*Table 1.* **Two Circles Toy Experiment.** We showcase the results from a single run. COH achieves 100% tuned features and 90% purity with $\varepsilon = 0.14$ coherence.

| MODEL | MSE | SPARSE% | MRL | %TUNED | MRL$_{180}$ | %TUN$_{180}$ | PURITY | %PURE | LOCALITY | COVERED |
|---|---|---|---|---|---|---|---|---|---|---|
| VANILLA | 0.0000996 | 83.3% | 0.44 | 43.0% | 0.14 | 0.0% | 0.17 | 0.0% | 0.53 | 0.44 |
| L1 | 0.0000995 | 28.8% | 0.50 | 52.0% | 0.22 | 0.0% | 0.31 | 1.2% | 4.62 | 4.58 |
| COH | 0.0000994 | 73.9% | 0.74 | 100.0% | 0.36 | 0.0% | 0.85 | 90.2% | 0.14 | 0.14 |

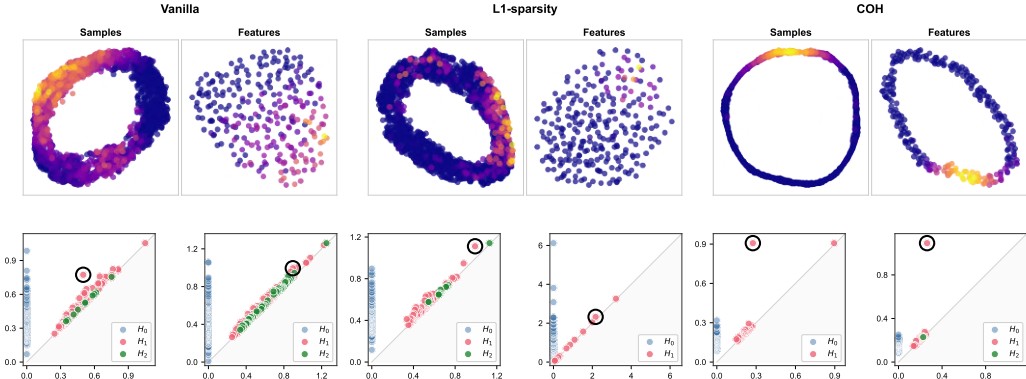

*Figure 4.* **Single Digit Experiment.** For a representative seed we show the UMAP projections of latent samples and latent features, with persistence diagrams for each. We highlight the single most persistent $H_1$ features, representing the expected circle. Samples are colored by activation of a single feature and features are colored by activation on a single sample. Only the COH model gives the expected circular topology in both spaces.

*Table 2.* **Single Digit Experiment** Results averaged over 5 random seeds. COH achieves 100% tuned features (MRL > 0.5) compared to 63% for L1, with lower variance across seeds.

| MODEL | MSE | SP% | MRL | %TUN | $MRL_{180}$ | $\%T_{180}$ | LOC | COV |
|---|---|---|---|---|---|---|---|---|
| VANILLA | 0.012±.000 | 53.3±7.9 | 0.46±.06 | 43.8±10.7 | 0.29±.05 | 13.2±8.1 | 1.11±.21 | 0.87±.13 |
| L1 | 0.010±.000 | 19.5±2.1 | 0.58±.01 | 63.3±2.8 | 0.43±.02 | 37.7±4.9 | 3.67±1.45 | 2.84±1.46 |
| COH | 0.011±.000 | 33.2±0.6 | 0.88±.00 | 100.0±0.0 | 0.63±.01 | 100.0±0.0 | 0.15±.00 | 0.15±.00 |

due to two qualitatively different solutions. Crucially, COH achieves what it is designed to do. The coherence metrics (Loc and Cov) are stable across all seeds with negligible variance. The algorithm reliably produces coherent latent spaces. The high variance in MRL and Purity does not reflect failure of the method. It reflects the fact that coherence admits multiple valid solutions. From Table 5 one can see that COH beats the other models in all interpretability metrics with the exception of MRL and %TUN score where $L^1$-sparsity wins. This is most likely due to sparseness. The high standard deviation for MRL, $MRL_{180}$ and Purity comes from the existence of two types of solutions. In some seeds the two digit classes stay separated. In others they merge into a single circle, losing labeling information but with excellent angular tuning. Both solutions are coherent. Both have interpretable features. They simply organize the data differently. In Figure 7, Figure 12 and Figure 11 we plot the UMAP embeddings, persistence diagrams and average samples over features for two of these different solutions. In Figure 7 one should pay attention to the persistent $H_0$ feature being present in the top latent space but not the bottom. This confirms the visual difference from UMAP. The top solution embeds the two digits in separate components (Figure 15) with angles looped around twice. The bottom solution merges the two circles (Figure 16) with no double loop. Both latent spaces are coherent as the UMAP projections and persistence diagrams confirm. The features in both cases attend to localized regions of the data manifold as intended.

To guide COH toward a specific coherent solution, we combine it with $L^1$ sparsity ($\lambda_{L^1} = 2 \times 10^{-2}$) and $\lambda_{COH} = 10^{-3}$. This biases optimization toward a disentangled solution, being the sparsest coherent representation, yielding reliable class separation and angle tuning across seeds while preserving coherence guarantees (see Figure 8, Figure 13 and Figure 14).

We again justify the 1-Lipschitz assumption by sampling $1000 \times 256$ pairs across ten seeds: $\psi$ satisfies the condition nearly exactly (violation rate < 0.2% in 9/10 seeds and last seed yields 1.46%); $\phi$ violates it on 3.4±2.0% of pairs with mean expansion $1.054 \pm 0.008$.

# 6. Token Embedding Experiment

In Large Language Models, words, parts of words and symbols are represented as token embeddings, that are learned during training. While these embeddings are typically signed, they can be made non-negative for the sake of interpretability. This is achieved through a non-negative activation function such as SOFTPLUS and guarantees a non-negative matrix $M$. It is well known that the token embedding space carries a meaningful geometry: e.g., "mother" is close to "father" and "March" is close to "February". Features, on the other hand, are often poly-semantic, and their space carries no clear geometry. We show that applying COH to the token embeddings yields interpretable features with a meaningful geometry. We do not claim, or hope, that *all* features become interpretable, as some are used to "merge" concepts. To demonstrate that the method works in this setting, we train a small BERT model (Devlin et al., 2019) on the WIKITEXT-2 dataset (Merity et al., 2017).

**Model.** We use a compact BERT-style encoder with a token embedding of dimension 256, 2 transformer blocks, and 4 transformer heads, trained on word-level WIKITEXT-2 with a vocabulary capped at the 2,000 most frequent tokens and a sequence length of 128. Inputs are tokenized at the word level (lowercased, with punctuation split off as separate tokens) rather than the more common subwords. This makes it easier to tell whether a feature is interpretable. Training follows the standard masked-language-modeling objective with a 15% masking rate and a label smoothing of 0.1, optimized with Adam using a learning rate of $3 \cdot 10^{-4}$ and a batch size of 128. We train for 300 epochs under a constant learning rate. The only non-standard choice is the token embedding, which we parametrize as a non-negative matrix using SOFTPLUS with $\beta = 20$.

We train three models with identical hyperparameters: one with the Vanilla (signed) token embedding, one with SOFTPLUS, and one with SOFTPLUS and COH. See Appendix B for further details on the COH model.

**Scoring interpretability and results.** We evaluate feature interpretability using three complementary methods. As a baseline, we treat the token geometry of the Vanilla model's embeddings as ground truth, since all models exhibit well-

structured token geometry.

For each feature, we examine its top 20 activating tokens and identify, within the Vanilla embedding, the token whose 20 nearest neighbors best match that activation set; we record this as **Mean Overlap**. Since overlap varies considerably across features, we additionally report the number of features achieving greater than $50\%$ overlap, which we record as **Overlap** $> .5$. See Figure 17 and Figure 18 for results.

We feed the top 10 activating tokens per feature into Claude Opus 4.7 (Anthropic, 2026), prompting it to make a binary judgment of whether the feature is interpretable, i.e., whether all 10 tokens share a common category; we record this as **Claude scoring**. See Table 8 for all interpretable features from the first seed and an explanation from Claude.

See Table 3 for a brief summary and Appendix B for more in-depth results. Model performance is negligibly impacted in our experiments; see Table 7.

| Metric | Coh | Softplus |
|---|---|---|
| Mean Overlap | $0.45 \pm 0.01$ | $0.22 \pm 0.00$ |
| Overlap $> .5$ | $77.4 \pm 3.3/256$ | $1.0 \pm 0.6/256$ |
| Claude Scoring | $87.6 \pm 10.4/256$ | $0.0 \pm 0.0/256$ |

*Table 3.* Mean and standard deviation for interpretability scores across 5 seeds.

As with the auto-encoders, coherence yields not only more interpretable features but a more interpretable feature *space*; see Figure 19 and Figure 20.

## 7. Discussion

Many approaches to interpretability focus on individual features. We instead focus on the interpretability of the feature space as a whole. We have given a rigorous framework for coupling latent samples and latent features so that the two share compatible topology, and we have shown that our objective reliably produces coherent representations: locality and covering are stable across seeds, and features inherit meaning from the samples they attend to.

The two settings validate this in complementary ways. In the autoencoders, where the data has known topology, coherence lets us verify the geometric guarantee directly, recovering the expected circles in both the sample and feature spaces. In the token embeddings no ground-truth topology exists; coherence instead transfers the embedding's known semantic geometry to the feature space. That the SOFT-PLUS-only baseline yields almost no interpretable features (on average 87.6 of 256 for COH versus none) shows that non-negativity alone is not enough, and the full feature listing lets the reader audit every judgment directly.

Our objective is simple and only requires a non-negative activation function, yet it carries across architecturally distinct setups. As a limitation, the Lipschitz assumption on the barycentric maps is verified empirically rather than enforced. A further observation, from the Double Digit Experiment, is that the same task can admit quite different coherent latent spaces, and our method does not control which one results. As COH $+ L^1$ demonstrates, coherence is complementary to other objectives: sparsity can select among coherent solutions without sacrificing the geometric guarantee.

For future work, we would like to scale to larger networks and datasets, apply coherence to tasks such as classification and to multiple layers rather than only the bottleneck or embedding layer, and explore disentanglement by applying the objective on blocks of the latent space or across multi-head architectures.

## Acknowledgments

We want to thank the anonymous reviewers, Chad Giusti and Daniela Egas Santander for constructive feedback. The work was supported by a grant from the Research Council of Norway (iMOD, NFR grant 325114); a Centre of Excellence grant (Centre for Algorithms in the Cortex, grant 332640) from the Research Council of Norway; and the Department of Mathematics at NTNU.

## Impact Statement

This paper presents work whose goal is to advance the field of Machine Learning, specifically interpretability of learned representations. We do not foresee direct negative societal consequences from this foundational research. If anything, improved interpretability may contribute positively to AI safety and transparency.

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

# A. Auto Encoder: Additional Plots and Tables.

**Algorithm**  Algorithm 1 presents the idealized coherence computation. In practice, several modifications are necessary for stable training:

**Scale normalization.**  When $M \in \mathbb{R}^{N \times L}$ with $N \neq L$, row and column distances live at different scales. We normalize by the mean pairwise distance (approximated by sampling) within each space:

$$\bar{d}_R = \frac{1}{\binom{N}{2}} \sum_{i<j} \|r_i - r_j\|_2, \quad \bar{d}_C = \frac{1}{\binom{L}{2}} \sum_{j<k} \|c_j - c_k\|_2.$$

The normalized variances become $\text{Var}_R(r_i)/\bar{d}_C^2$ and $\text{Var}_C(c_j)/\bar{d}_R^2$ and the normalized coverings become $\text{Cov}_R(r_i)/\bar{d}_R^2$ and $\text{Cov}_C(c_j)/\bar{d}_C^2$.

**Top-$k$ aggregation.**  Rather than penalizing mean variance, we penalize the $k$ worst offenders, making the loss robust to outliers, moreover the $\epsilon$-coherence is based on the worst offender:

$$\mathcal{L}_{\text{var}} = \frac{1}{k} \sum_{i \in \text{top-}k} \text{Var}_R(r_i) + \frac{1}{k} \sum_{j \in \text{top-}k} \text{Var}_C(c_j).$$

**Target variance/covered threshold.**  Perfect coherence ($\epsilon = 0$) is unnecessarily restrictive, only possible for a matrix of constant orthogonal blocks. We thus only penalize variance exceeding a target parameter $\tau$:

$$[\text{Var}(x) - \tau]_+ = \max(0, \text{Var}(x) - \tau)$$

and

$$[\text{Cov}(x) - \tau]_+ = \max(0, \text{Cov}(x) - \tau).$$

**Normalization kernel.**  We choose squared $L^1$ normalization as our normalization kernel, that is

$$w_j^{(i)} = \frac{M_{ij}^2}{\sum_k M_{ik}^2} \quad \text{and} \quad v_i^{(j)} = \frac{M_{ij}^2}{\sum_k M_{kj}^2}.$$

This concentrates weight on dominant activations without introducing a temperature hyperparameter.

See Algorithm 1 for pseudocode. Variance simplifies via $Var(X) = \mathbb{E}[X^2] - \mathbb{E}[X]^2$:

$$\text{Var}_R(r_i) = \sum_j W_{ij} \|c_j\|^2 - \|\phi(r_i)\|^2.$$

Covering terms expand similarly. Complexity is $\mathcal{O}(B^2 L + BL^2)$ per batch, dominated by barycenter computation.

**Hyperparameter selection.**  For the single-digit experiment, COH performs well across a range of $\lambda$ values while maintaining stable MSE. Although L1 achieves slightly lower reconstruction error, COH produces better-tuned features even with less sparsity. We select $\lambda_{\text{COH}} = \lambda_{L^1} = 10^{-3}$.

For the two-digit experiment, we observe angular tuning modulo 180 degrees rather than full 360 degrees tuning. As $\lambda_{\text{COH}}$ increases, $\text{MRL}_{180}$ improves, but purity drops sharply at $\lambda = 5 \times 10^{-3}$ where the two latent circles merge, collapsing class information while preserving angular structure. We select $\lambda_{\text{COH}} = \lambda_{L^1} = 10^{-3}$.

**Decoding angles.**  Using circular coordinates (de Silva et al., 2011) (implementation from (Perea et al., 2023)), we extract angles from the most persistent $H_1$ class in both the latent sample and feature spaces. Figures 10 and 15 show these recovered angles plotted against the true generating angles. To transfer angles from features to samples, we assign each sample the angle of its maximally-activating feature. The strong agreement confirms that the circle discovered in feature space corresponds to the circle in sample space, demonstrating how coherent features enable direct readout of latent structure.

*Table 4.* **Hyperparameter sweep on rotated MNIST.** Sp.% = mean active features per sample (active if $> 1\%$ of max). T% = fraction of features with MRL $> 0.5$. P% = fraction of features with component score $> 0.5$. Bold rows indicate selected hyperparameters. The lower MSE for the two digit experiment in general is due to doubling the sample set.

| | | **SINGLE DIGIT** | | | | | | **TWO DIGITS** | | | | | |
| --- | --- | --- | --- | --- | --- | --- | --- | --- | --- | --- | --- | --- | --- |
| METHOD | $\lambda$ | MSE | Sp.% | MRL | %T | METHOD | $\lambda$ | MSE | Sp.% | MRL | MRL$_{180}$ | PUR. | %P |
| VAN. | – | .011 | 38 | .55 | 58 | VAN. | – | .009 | 44 | .38 | .38 | .40 | 34 |
| COH | $10^{-5}$ | .011 | 46 | .48 | 49 | COH | $10^{-5}$ | .008 | 53 | .34 | .34 | .35 | 27 |
| COH | $10^{-4}$ | .011 | 51 | .58 | 69 | COH | $10^{-4}$ | .008 | 56 | .39 | .41 | .32 | 20 |
| COH | $5\times10^{-4}$ | .012 | 36 | .84 | 100 | COH | $5\times10^{-4}$ | .009 | 41 | .47 | .62 | .61 | 79 |
| **COH** | $\mathbf{10^{-3}}$ | **.012** | **33** | **.88** | **100** | **COH** | $\mathbf{10^{-3}}$ | **.009** | **41** | **.24** | **.51** | **.88** | **97** |
| COH | $5\times10^{-3}$ | .011 | 32 | .91 | 100 | COH | $5\times10^{-3}$ | .009 | 40 | .15 | .74 | .10 | 0 |
| COH | $10^{-2}$ | .012 | 30 | .92 | 100 | COH | $10^{-2}$ | .009 | 38 | .12 | .76 | .10 | 0 |
| L1 | $10^{-5}$ | .012 | 48 | .50 | 49 | L1 | $10^{-5}$ | .009 | 48 | .34 | .35 | .37 | 30 |
| L1 | $10^{-4}$ | .011 | 36 | .58 | 66 | L1 | $10^{-4}$ | .009 | 44 | .35 | .36 | .34 | 28 |
| L1 | $5\times10^{-4}$ | .010 | 27 | .60 | 68 | L1 | $5\times10^{-4}$ | .008 | 34 | .39 | .37 | .33 | 23 |
| **L1** | $\mathbf{10^{-3}}$ | **.010** | **23** | **.56** | **60** | **L1** | $\mathbf{10^{-3}}$ | **.007** | **29** | **.42** | **.37** | **.37** | **31** |
| L1 | $5\times10^{-3}$ | .011 | 10 | .54 | 60 | L1 | $5\times10^{-3}$ | .008 | 18 | .41 | .36 | .35 | 27 |
| L1 | $10^{-2}$ | .013 | 7 | .51 | 43 | L1 | $10^{-2}$ | .012 | 9 | .46 | .42 | .54 | 57 |

*Table 5.* **Double Digit Experiment.** Results averaged over 10 random seeds. Both digits lack 180 degree symmetry but high MRL$_{180}$ scores reflect the tendency of networks to collapse antipodal angles. COH achieves better angular tuning (180), with almost 2 times higher purity and at modest reconstruction cost. We do however have high variance over interpretability metrics over seeds, however, COH achieves consistent coherence (Loc, Cov) across all seeds. Variance in purity reflects distinct but equally coherent solutions (see Figure 7, Figure 11, Figure 12). COH + L1 combines both objective functions, guides COH towards a sparse solution and reliably achieves class separation and angle tuning while maintaining coherence, at the cost of higher MSE.

| MODEL | MSE | Sp% | MRL | %TUN | MRL$_{180}$ | %T$_{180}$ | PUR | %PUR | LOC | COV |
| --- | --- | --- | --- | --- | --- | --- | --- | --- | --- | --- |
| VANILLA | 0.009±.000 | 54.8±3.1 | 0.33±.01 | 19.6±2.1 | 0.32±.02 | 17.0±3.9 | 0.33±.03 | 6.2±2.3 | 0.64±.04 | 0.53±.02 |
| L1 | 0.008±.000 | 28.0±3.0 | 0.42±.02 | 35.3±4.7 | 0.38±.01 | 29.3±3.5 | 0.37±.02 | 8.8±0.9 | 1.08±.15 | 0.74±.09 |
| COH | 0.009±.000 | 38.6±1.6 | 0.34±.17 | 26.2±26.7 | 0.65±.06 | 88.4±11.2 | 0.61±.27 | 44.5±36.0 | 0.16±.01 | 0.16±.01 |
| COH + L1 | 0.014±.000 | 17.8±1.8 | 0.85±.01 | 99.0±0.9 | 0.70±.01 | 98.7±0.9 | 0.90±.01 | 91.9±2.1 | 0.15±.00 | 0.15±.00 |

*Table 6.* **Hyperparameters used in all experiments.** We choose non-negative activation function, Softplus, with a high $\beta$ value to encourage sparsity while mitigating the 'dead neuron' issue typically associated with ReLU. We use AdamW optimizer with $lr = 10^{-3}$, weight decay $10^{-5}$, cosine annealing schedule (eta$_{\min}$ = $lr/10$), and gradient clipping at norm 1.0.

| Hyperparameter | Value |
|---|---|
| Hidden dim | 512 |
| Latent dim | 256 |
| Encoder/Decoder layers | 5/5 |
| Activation hidden | GELU |
| Activation latent | Softplus($\beta = 20$) |
| Learning rate | $10^{-3}$ |
| Batch size | 1024 |
| Epochs | 300 |
| $\lambda_{\text{COH}}$ | $10^{-3}$ |
| $\lambda_{\text{L1}}$ | $10^{-3}$ |
| Target variance/covering ($\tau$) | 0.1 |
| Top-$k$ (rows/cols) | 15/15 |

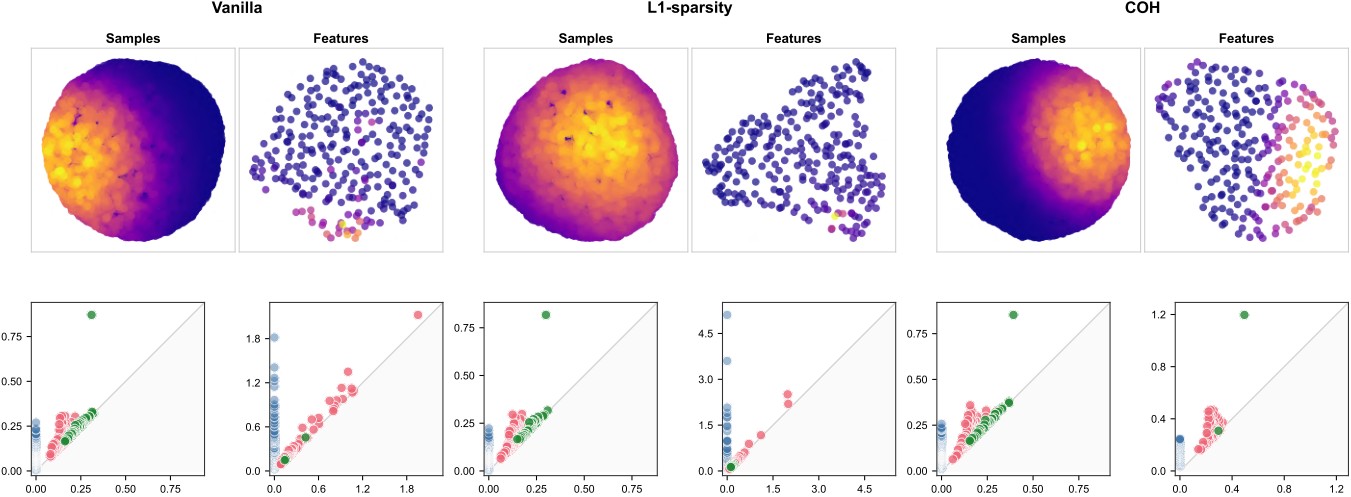

*Figure 5.* **Toy experiment: sphere.** We replicate the two circle toy experiment with a sphere.

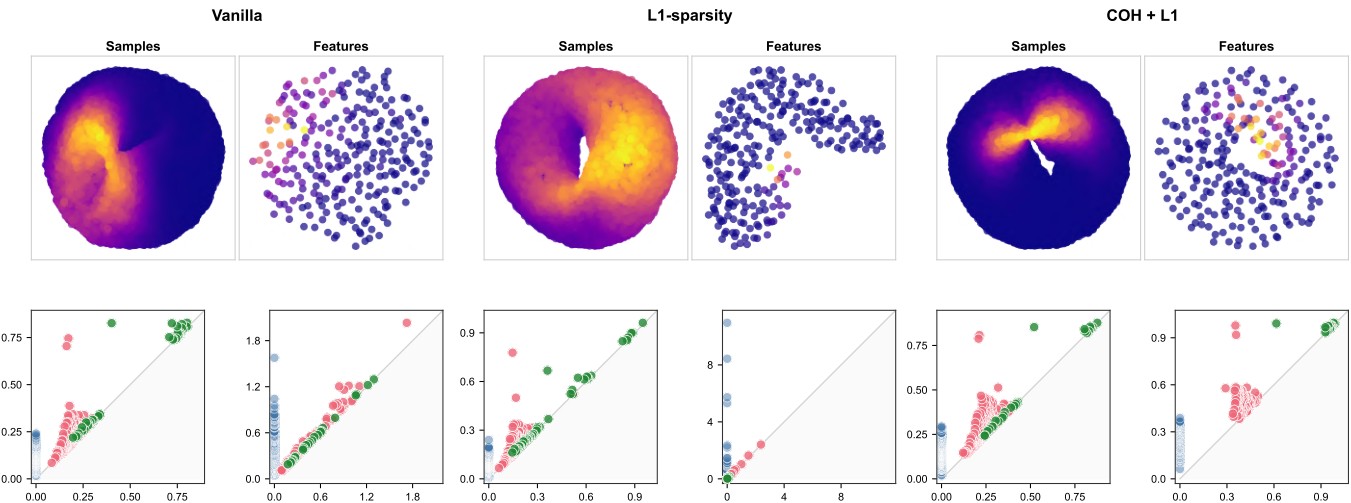

*Figure 6.* **Toy experiment: torus.** We replicate the two circle toy experiment with a torus.

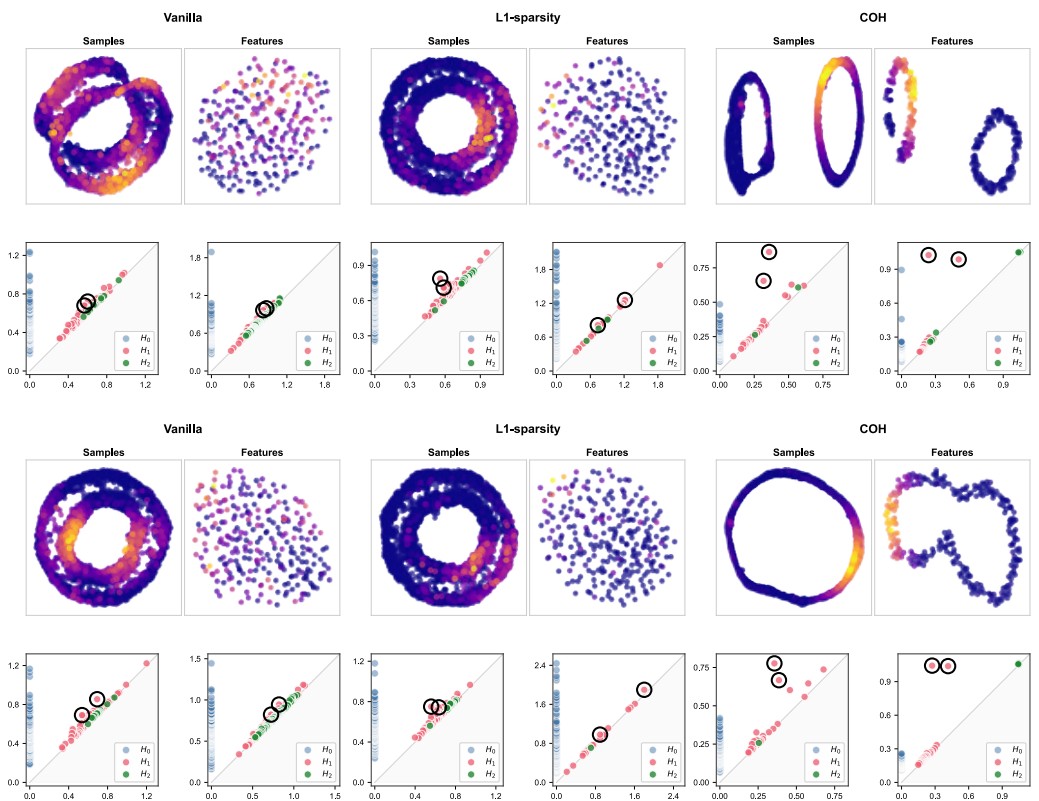

*Figure 7.* **Double Digit Experiment.** For two different seeds we show UMAP projections of latent samples and latent features, with persistence diagrams for each. We highlight the two most persistent $H_1$ features, representing the two expected circles. Samples are colored by activation of a single feature and features are colored by activation on a single sample. The seeds are picked as to show the diversity in the learned latent spaces.

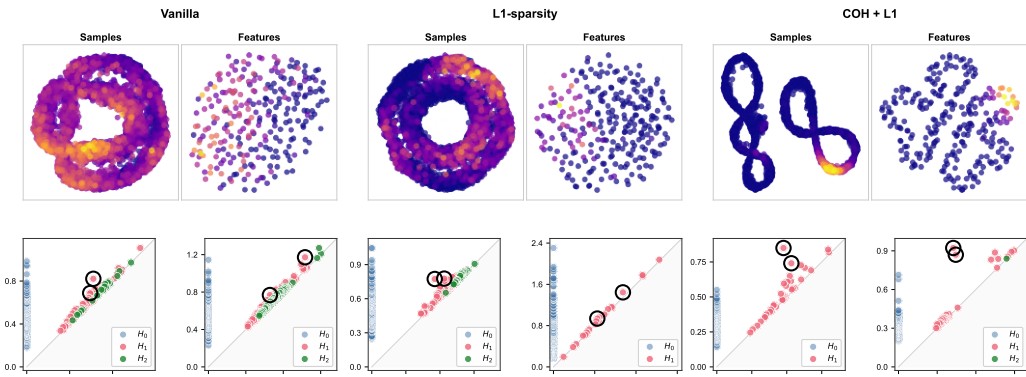

*Figure 8.* **Double Digit Experiment (COH + L1).** For two different seeds we show UMAP projections of latent samples and latent features, with persistence diagrams for each. We highlight the two most persistent $H_1$ features, representing the two expected circles. Samples are colored by activation of a single feature and features are colored by activation on a single sample.

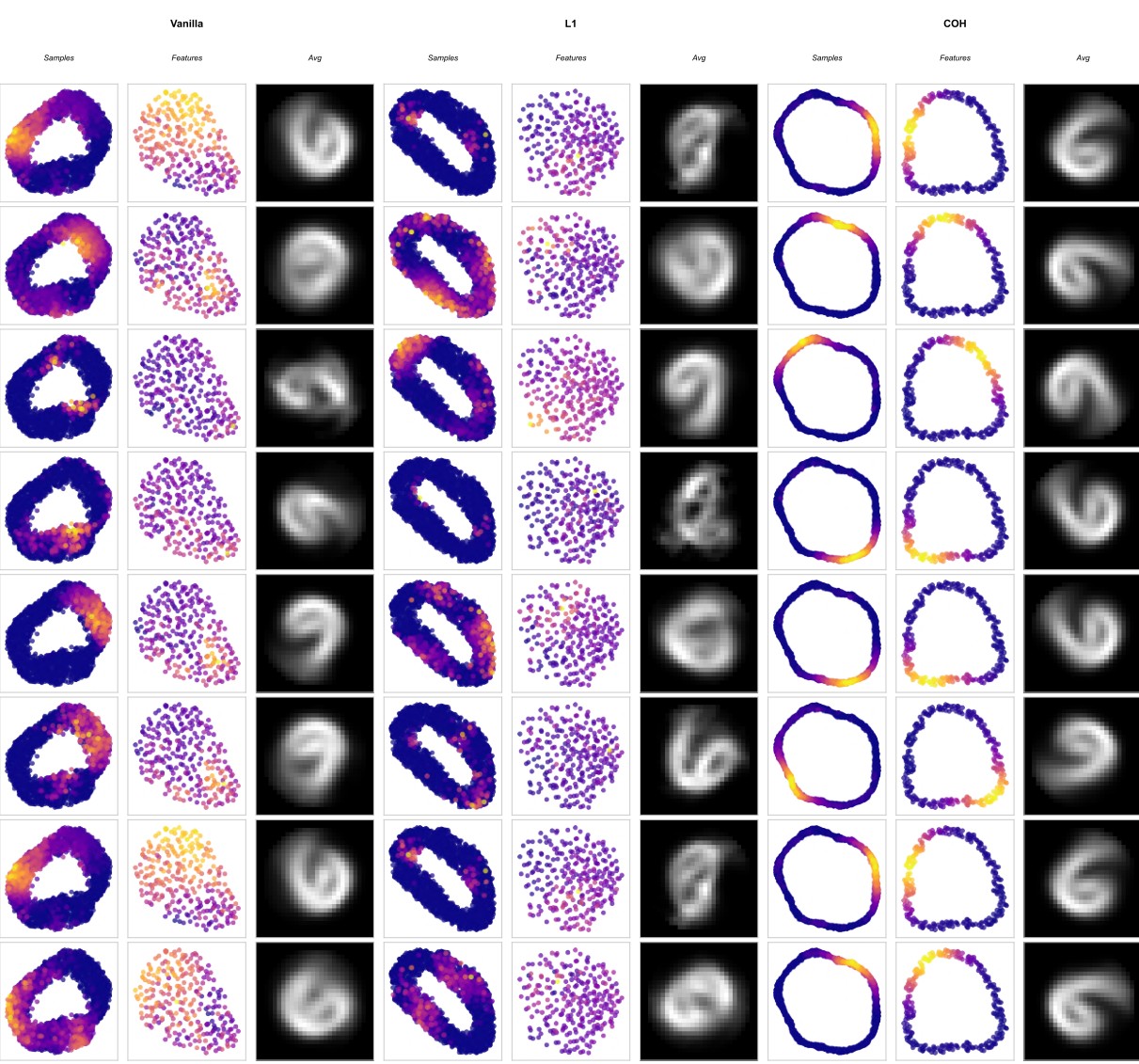

*Figure 9.* **Single digit experiment.** We plot UMAP projections of latent samples and latent features colored by a latent feature and sample representatively. For a random sample of features we plot the weighted sum of the original images, the random features corresponds to the coloring of the latent samples.

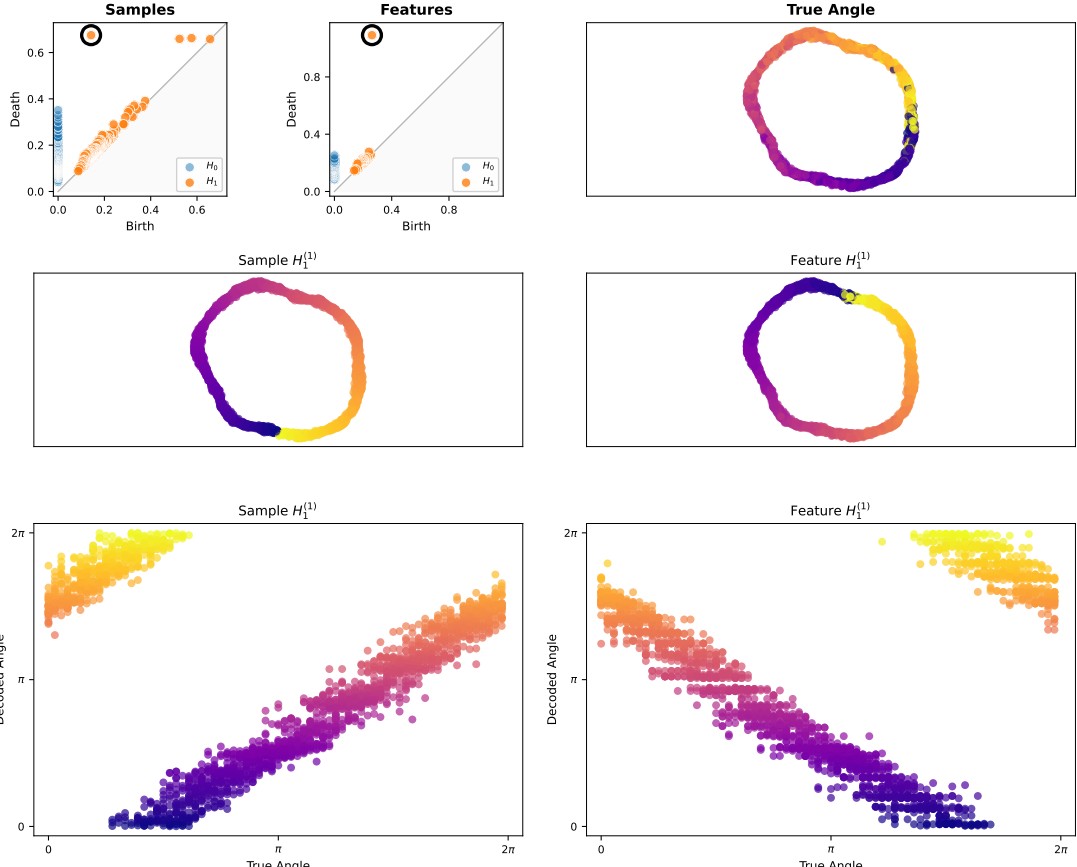

*Figure 10.* **Single digit experiment.** We analyse at a representative latent space from the single digit experiment using circular coordinates from persistence (co)homology, and compare it against the true generating angles. Second row we color the latent samples by circular coordinates. In the last row we plot circular coordinates against the true angle.

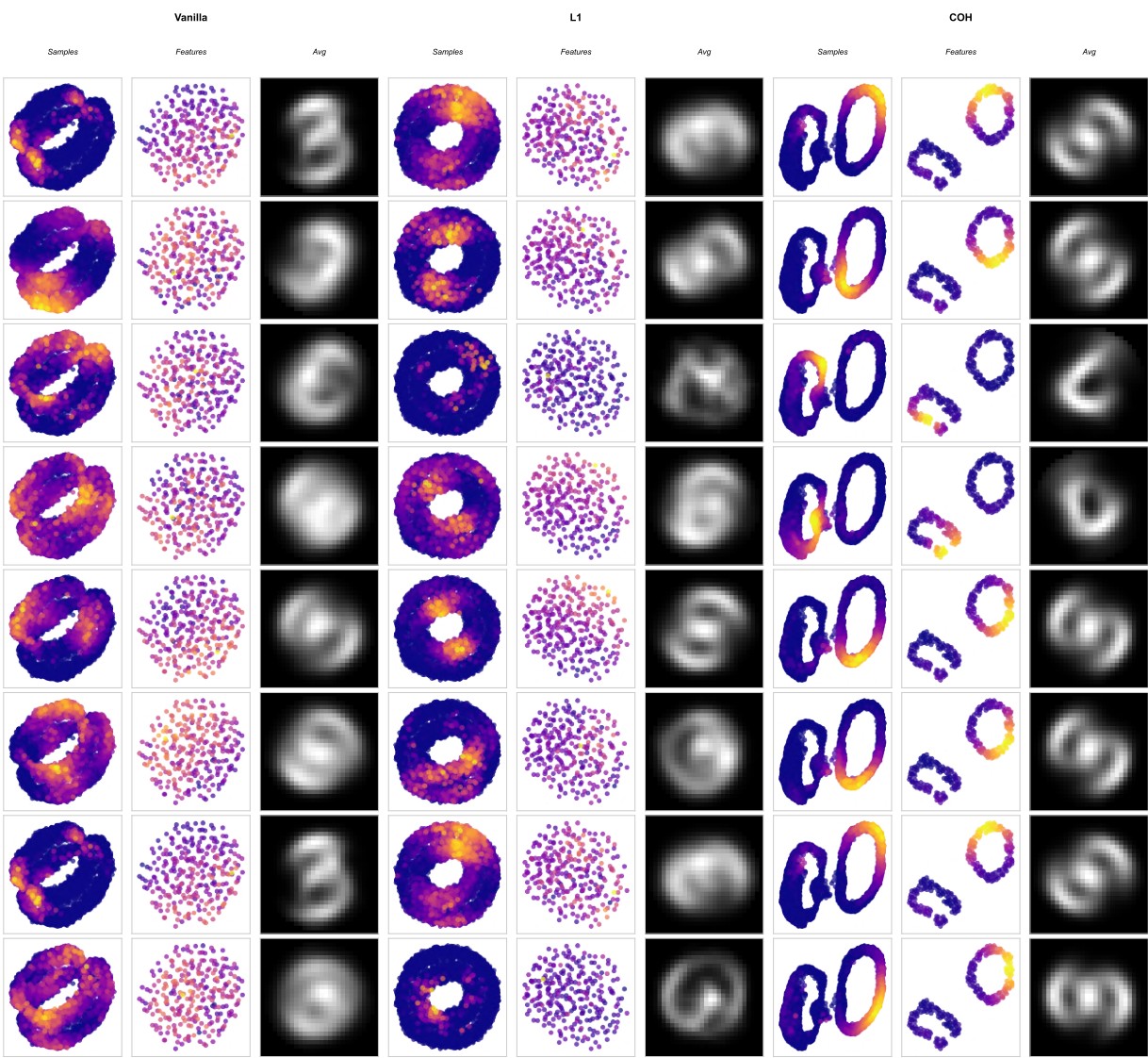

*Figure 11.* **Double digit experiment (Separated).** We plot UMAP projections of latent samples and latent features colored by a latent feature and sample representatively. For a random sample of features we plot the weighted sum of the original images, the random features corresponds to the coloring of the latent samples. We note COH nicely distinguishes the two classes into two circles. Features corresponding to the label 3, have angular tuning modulo 180 degrees, while features corresponding to the digit 7 have a 360 degree angular tuning.

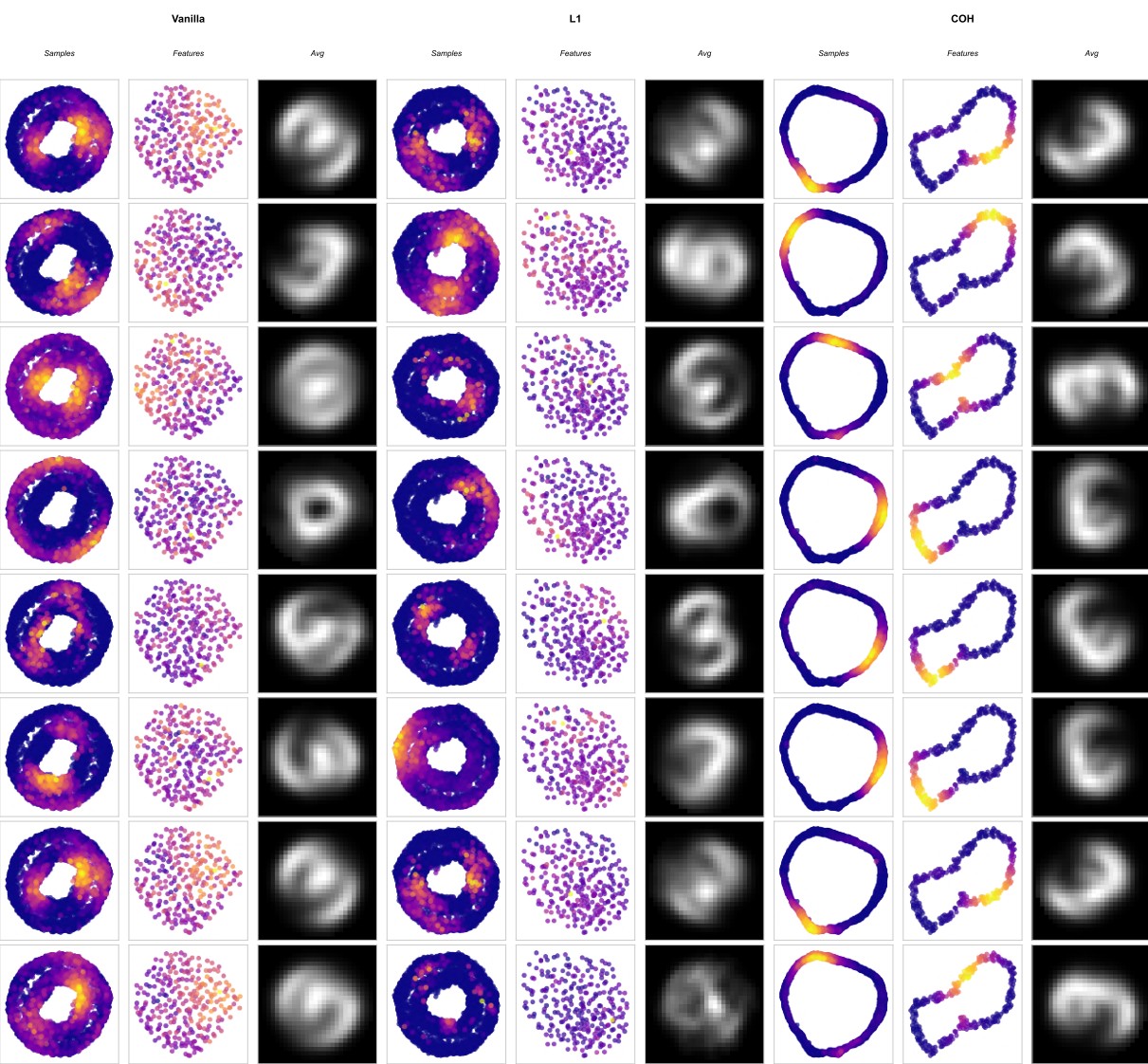

*Figure 12.* **Double digit experiment (Merged).** We plot UMAP projections of latent samples and latent features colored by a latent feature and sample representatively. For a random sample of features we plot the weighted sum of the original images, the random features corresponds to the coloring of the latent samples. We note, in this case, COH has merged the two classes into the same circle, but have excellent angular tuning.

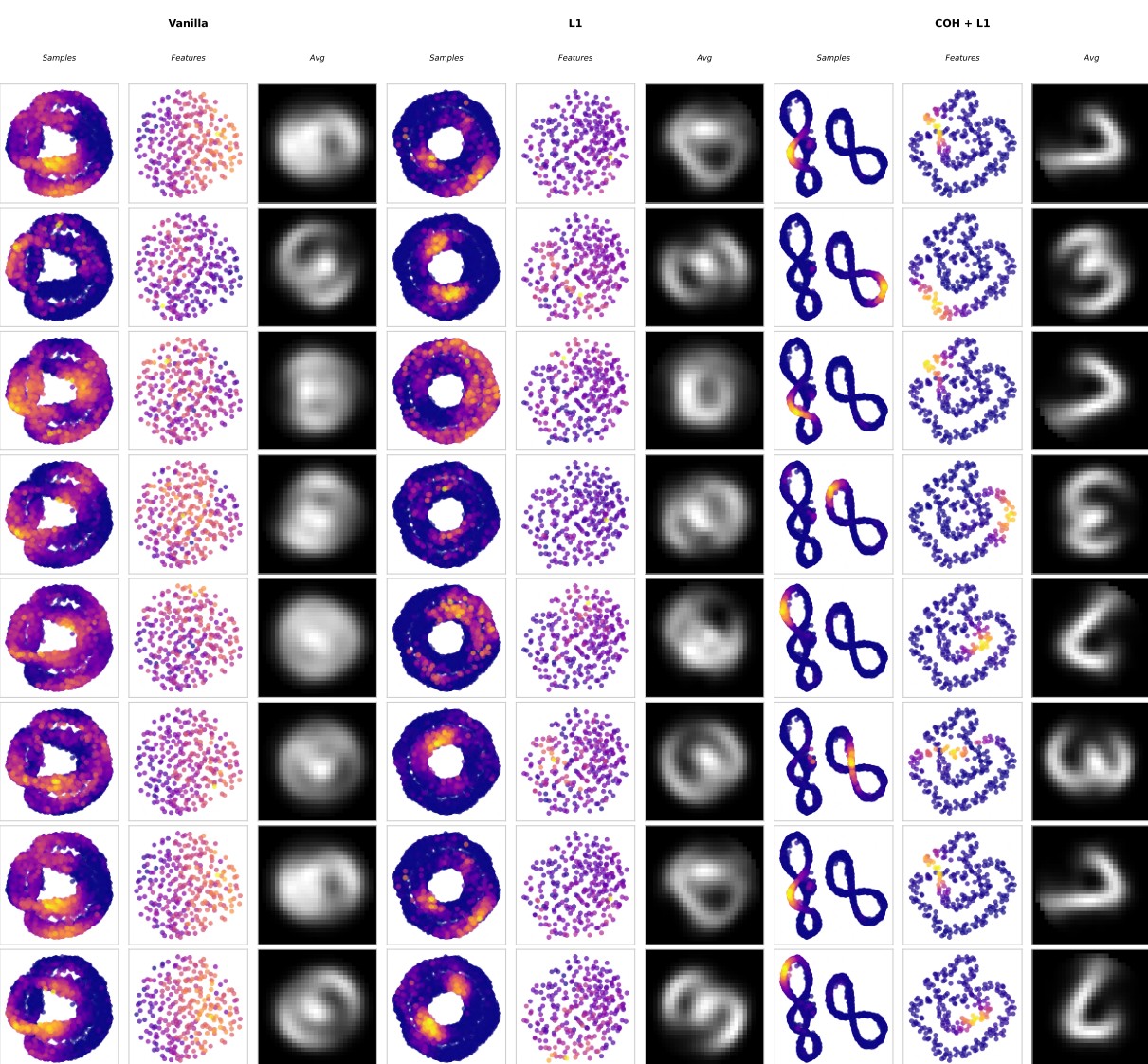

*Figure 13.* **Double digit experiment ($L^1$ and COH) .** We plot UMAP projections of latent samples and latent features colored by a latent feature and sample representatively. For a random sample of features we plot the weighted sum of the original images, the random features corresponds to the coloring of the latent samples. Using COH and L1 together we get a much cleaner results.

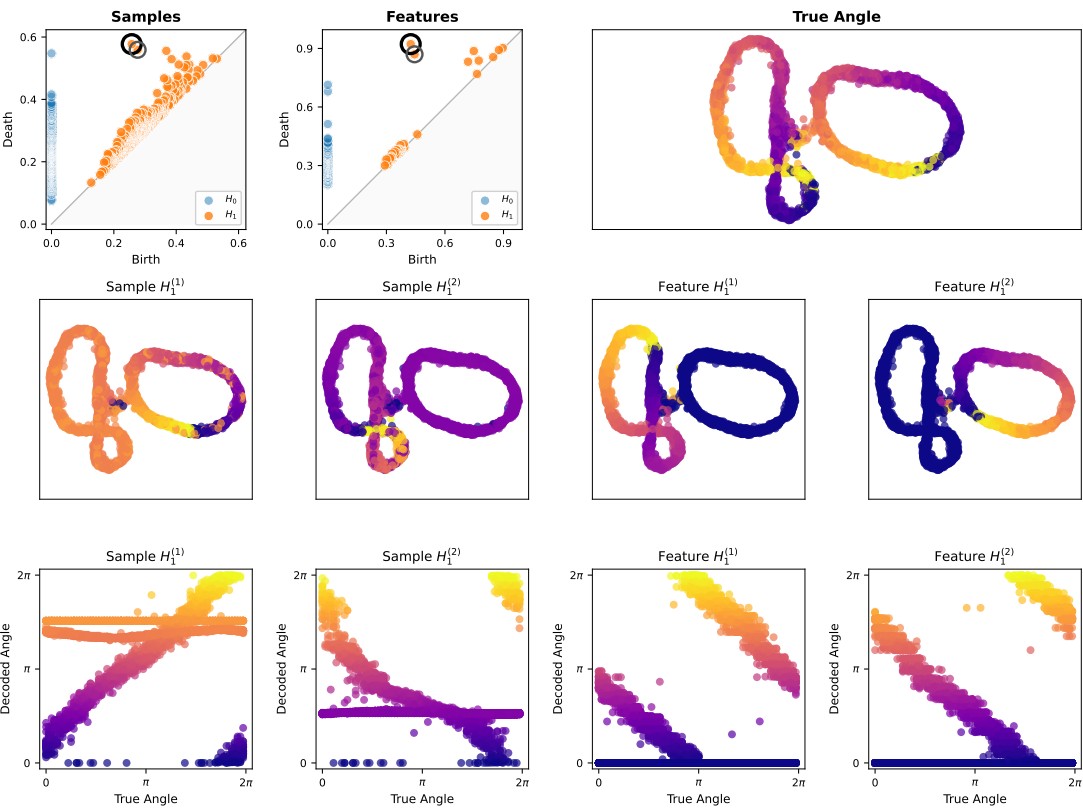

*Figure 14.* **Double digit experiment** ($L^1$ **and** COH**).** We find a separated a latent space in the double digit experiment using circular coordinates from persistence (co)homology, and compare it against the true generating angles. Note that we have both good angle tuning and component tuning of the features.

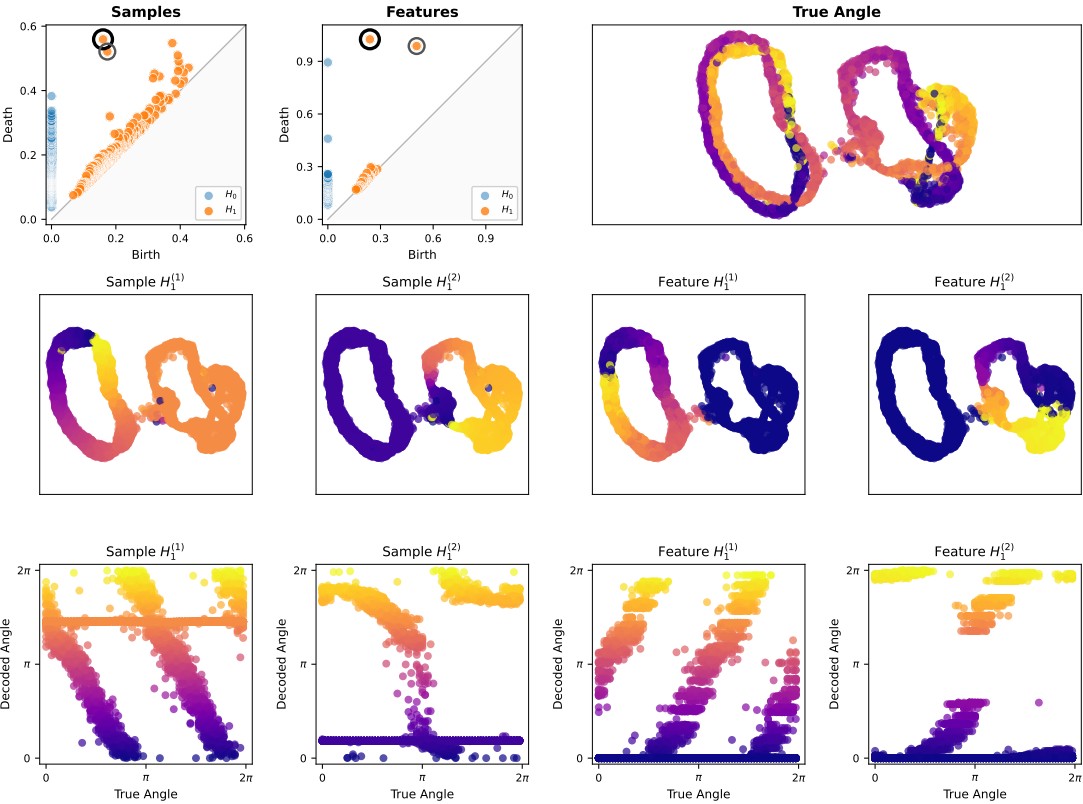

*Figure 15.* **Double digit experiment (Separated).** We find a separated latent space in the double digit experiment using circular coordinates from persistence (co)homology, and compare it against the true generating angles.

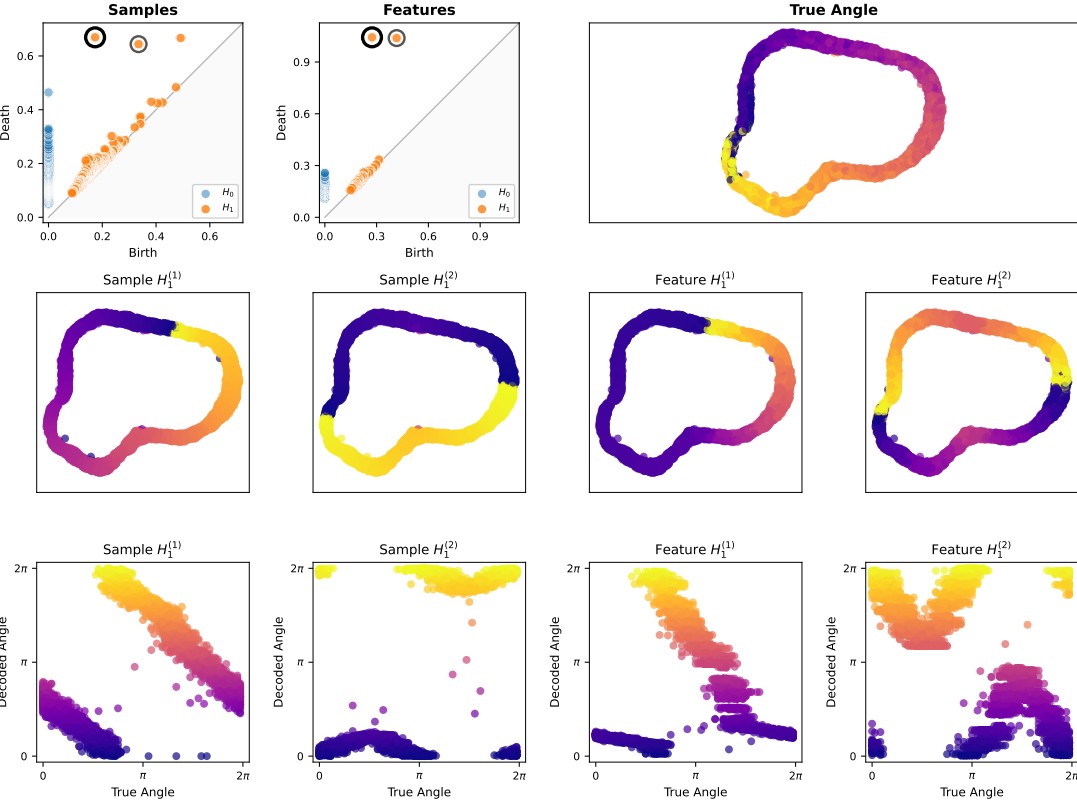

*Figure 16.* **Double digit experiment (Merged).** We analyse here a merged latent space from the double digit experiment using circular coordinates from persistence (co)homology, and compare it against the true generating angles.

# B. BERT: Additional Plots and Tables.

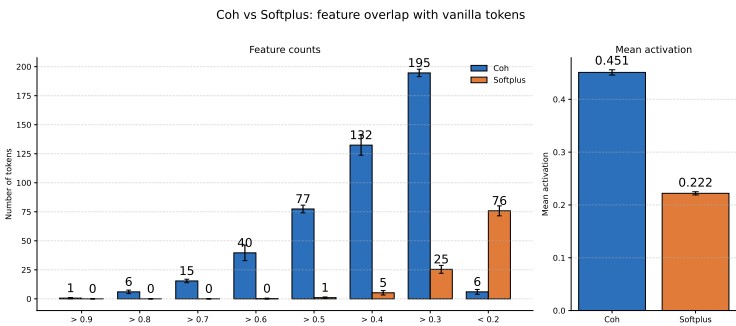

*Figure 17.* Mean $\pm$ std over 5 seeds for the overlap of the top 20 tokens per feature against the 20-nearest-neighbor token neighborhood in the Vanilla embedding.

**Scoring interpretability using LLMs.**   For the first seed of each of the two non-negative token embeddings, we examine the top 10 tokens for each of the 256 features. We feed this list of features into Claude Opus 4.7 and ask it to assign a binary score to each feature according to whether it is interpretable. It finds 81 "interpretable" features in the COH embedding and zero in the SOFTPLUS embedding. We list all of these features, represented by their top 10 tokens, together with an explanation, in Table 8. We note that while the SOFTPLUS embedding has zero pure features, it has some that come close, such as ['scotland', 'moving', 'australia', 'wales', 'cardiff', 'ireland', 'virginia', 'located', 'india', 'pennsylvania'], which seems to lean toward locations; but if we were to count this loosely, the COH model would have nearly all of its features classified as "interpretable". We also note that the number of features Claude judges interpretable is similar to the count given by the overlap measure, but the actual numbers by the LLM judge should, of course, be taken with a grain of salt.

**Implementation details.**   We note that COH can "cheat" by separating the token embedding into two clusters, such as verbs and the rest of the vocabulary. The loss can then push these clusters apart, increasing the row and column scales and thereby decreasing the coherence score. There are several ways to address this; the one we use here is to sample *locally*. We begin by sampling the rows of the matrix globally (at random), and as the embedding becomes more coherent, we sample sub-matrices locally, based on distance anchors chosen at random.

A second issue is that the coherence loss can push tokens to fit the topology of the features, whereas we would rather it push features to fit the topology of the tokens. We encourage this by downscaling the terms governing row variance and covering by a factor of 10 relative to their column counterparts.

We run one "global" iteration of the coherence objective, as in 1, per batch, sampling 1024 tokens and using all features. Simultaneously, we run 5 "local" iterations: from the 1024 tokens we pick 5 anchors at random, and from each anchor we form a sub-matrix of dimension $128 \times 32$ based on the 128 closest tokens to the anchor and the 32 most active features given those tokens. We gradually introduce the local sub-matrices using a scheduler based on the coherence of the previous epoch.

| Method | Acc | Coherence | Time/epoch |
|---|---|---|---|
| Vanilla | $0.5290 \pm 0.003$ | – | $1.85s$ |
| Softplus | $0.5329 \pm 0.002$ | $0.5008 \pm 0.0021$ | $1.87s$ |
| Coh | $0.5236 \pm 0.004$ | $0.2072 \pm 0.0261$ | $3.68s$ |

*Table 7.* Average results over 5 seeds after 300 epochs.

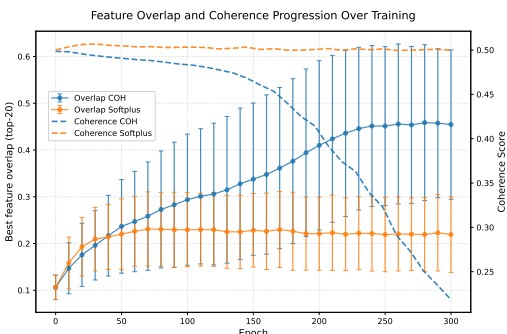

*Figure 18.* For the first seed, we plot the average best feature overlap with a token neighborhood of the Vanilla model at each epoch. We also track the average coherence score per epoch.

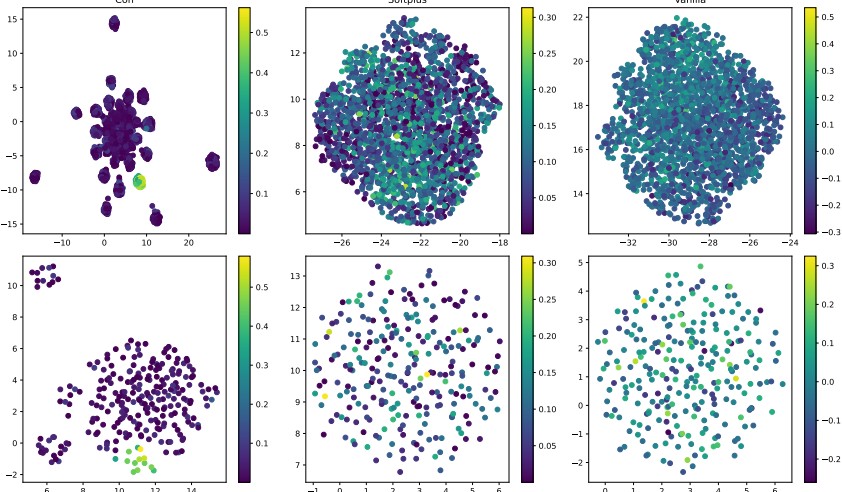

*Figure 19.* UMAP projections of the token and feature embeddings for the first seed. The plots are colored by the values of a single token vector and a single feature vector.

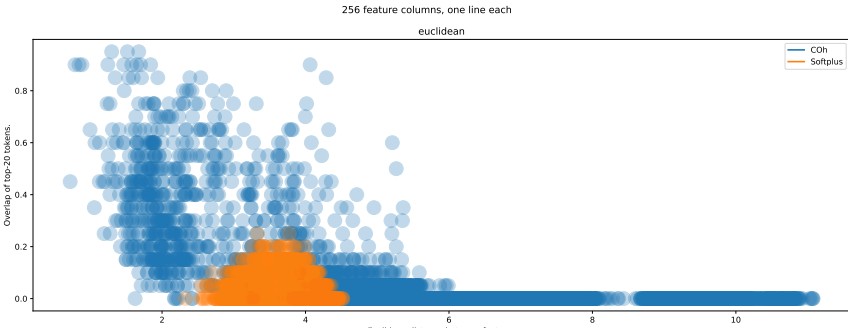

*Figure 20.* Again for the first seed: for each feature we plot, on the $x$-axis, its distance to its 20 nearest feature vectors, and on the $y$-axis, the overlap of its top 20 tokens with those of the neighboring feature. In words, each point answers: how far am I from that feature ($x$-axis), and how similar am I to it in terms of top-20 tokens ($y$-axis)? This showcases that the coherent embedding has more meaningful geometry for the features.

*Table 8.* A list of all "pure" features according to Claude for the first seed for the coherent embedding.

| Feat. | Top 10 tokens | Concept |
|---|---|---|
| *Numbers* | | |
| 4 | 50, ten, 24, 100, 60, 20, 19, 15, 40, 12 | Cardinal numbers |
| 29 | 11, 13, 17, 23, 18, 14, 24, 22, 38, 26 | Two-digit numbers |
| 42 | 30, 45, 39, 29, 33, 28, 32, 50, 70, 19 | Two-digit numbers |
| 136 | 26, 13, 18, 28, 31, 27, 17, 23, 39, 22 | Two-digit numbers |
| 143 | 14, 13, 12, 22, 11, 24, 28, 19, 16, 17 | Two-digit numbers |
| 144 | 75, 45, 38, 8, 70, 80, 40, 31, 36, 35 | Two-digit numbers |
| 146 | eight, ten, nine, seven, 300, 24, 18, 14, 23, 27 | Numbers |
| 177 | 28, 27, 80, 33, seven, 35, 31, 39, 60, 90 | Numbers |
| 186 | 400, 200, 16, 150, 100, 26, 23, 300, 24, 19 | Numbers |
| 220 | 75, 80, 90, 60, 100, 38, 300, 200, 70, 50 | Numbers |
| 239 | 29, 28, 75, 24, 14, 11, 27, 22, 45, 21 | Two-digit numbers |
| *Years* | | |
| 10 | 2007, 1972, 2005, 1999, 1984, 1970, 1964, 2016, 2008, 1960 | Years |
| 26 | 1999, 2014, 1997, 2002, 1994, 1991, 1964, 2015, 1996, 2004 | Years |
| 36 | 1993, 2002, 1997, 1999, 1986, 2003, 1985, 1984, 1994, 1992 | Years |
| 112 | 2009, 2014, 2008, 2010, 2007, 2013, 2011, 2012, 2015, 2004 | Years |
| 165 | 1998, 1991, 1995, 1999, 1993, 1994, 1972, 1997, 1986, 1990 | Years |
| 190 | 2016, 2015, 2014, 2011, 1940, 2013, 2004, 2009, 1999, 2010 | Years |
| 241 | 1993, 1988, 1998, 1986, 1970, 1984, 1997, 2002, 1989, 2001 | Years |
| 252 | 2005, 2001, 2002, 2004, 2006, 1995, 2000, 1992, 2003, 1988 | Years |
| *Verbs by semantic role* | | |
| 0 | entered, killed, attacked, struck, arrived, moved, captured, formed, returned, ran | Motion / action past verbs |
| 65 | turned, sold, cut, passed, dropped, defeated, reached, entered, divided, struck | Past action verbs |
| 72 | give, see, write, know, bring, follow, keep, want, begin, make | Basic action verbs |
| 81 | travel, meet, hold, find, produce, reach, stop, build, fight, follow | Action verbs |
| 93 | find, write, produce, create, believe, build, know, hold, reach, leave | Create / produce verbs |
| 189 | leave, stop, get, meet, follow, keep, find, allow, continue, try | Continuation verbs |
| 245 | avoid, follow, keep, prevent, give, provide, begin, continue, perform, build | Continuation / prevention |
| 54 | established, rejected, introduced, accepted, developed, founded, sold, played, abandoned, controlled | Institutional action verbs |

*Table 8.* A list of all "pure" features according to Claude for the first seed for the coherent embedding.

| Feat. | Top 10 tokens | Concept |
|---|---|---|
| 57 | performed, played, produced, recorded, joined, met, served, done, created, achieved | Career / creative verbs |
| 61 | released, raised, removed, defeated, adopted, performed, achieved, published, gained, left | Achievement past verbs |
| 89 | provided, held, faced, created, shot, ran, introduced, sent, fired, paid | Past-tense action verbs |
| 97 | extended, brought, left, adopted, passed, represented, earned, introduced, broken, turned | Past action verbs |
| 127 | formed, developed, sustained, struck, affected, taken, controlled, damaged, occurred, created | Damage / formation verbs |
| 128 | opened, signed, started, attended, earned, founded, constructed, married, met, issued | Life-event verbs |
| 155 | took, opened, grew, ran, brought, adopted, carried, taken, turned, started | Past action verbs |
| 175 | maintained, accepted, held, remained, created, developed, discovered, passed, visited, formed | Maintain / hold verbs |
| 179 | published, issued, launched, conducted, provided, established, featured, included, founded, visited | Publish / launch verbs |
| 185 | seen, allowed, kept, used, observed, offered, lived, inspired, ordered, determined | Past participle verbs |
| 187 | reduced, killed, affected, damaged, raised, sustained, increased, destroyed, controlled, carried | Damage / change verbs |
| 193 | appointed, promoted, recognized, awarded, listed, elected, assigned, ordered, proposed, named | Appointment / award verbs |
| 210 | continued, intended, used, described, peaked, found, refused, lived, managed, allowed | Past verbs |
| 211 | faced, affected, represented, damaged, covered, controlled, captured, destroyed, defeated, occurred | Affect / damage verbs |
| 212 | adopted, signed, replaced, marked, attacked, joined, achieved, rejected, earned, paid | Action past verbs |
| 223 | dropped, planned, made, gained, produced, shot, maintained, raised, arrived, captured | Past action verbs |
| 229 | produced, developed, issued, caused, struck, shot, completed, recorded, marked, formed | Produce / cause verbs |
| 235 | lost, achieved, filmed, dropped, held, produced, discovered, entered, met, faced | Past action verbs |

*Speech / cognition verbs*

| Feat. | Top 10 tokens | Concept |
|---|---|---|
| 66 | expressed, claimed, concluded, felt, revealed, attempted, decided, agreed, explained, meant | Cognition / statement |
| 121 | commented, wanted, noted, appeared, agreed, read, praised, wrote, criticized, got | Reaction verbs |

*Table 8.* A list of all "pure" features according to Claude for the first seed for the coherent embedding.

| Feat. | Top 10 tokens | Concept |
| --- | --- | --- |
| 166 | claimed, wanted, argued, suggested, saw, appear, felt, stated, attempted, read | Cognition / statement |
| 176 | revealed, showed, read, concluded, meant, uses, suggested, got, claimed, felt | Reveal / conclude verbs |
| 232 | noted, suggested, thought, meant, argued, revealed, showed, believed, expressed, concluded | Thought / argument verbs |

*Verb participles (-ing)*

| Feat. | Top 10 tokens | Concept |
| --- | --- | --- |
| 111 | losing, remaining, growing, returning, winning, resulting, doing, selling, coming, finding | -ing participles |

*Adverbs by function*

| Feat. | Top 10 tokens | Concept |
| --- | --- | --- |
| 87 | typically, thus, always, usually, finally, therefore, ultimately, probably, initially, yet | Sentence adverbs |
| 94 | probably, initially, still, sometimes, finally, therefore, once, eventually, now, thus | Temporal adverbs |
| 124 | once, even, immediately, sometimes, finally, possibly, previously, almost, initially, soon | Temporal adverbs |
| 150 | now, actually, already, nearly, still, almost, always, typically, simply, sometimes | Temporal / degree |
| 170 | currently, particularly, especially, usually, eventually, always, thus, often, simply, perhaps | Sentence adverbs |
| 217 | significantly, typically, generally, mainly, slightly, primarily, officially, fully, relatively, initially | Degree / manner |
| 238 | possibly, mainly, particularly, mostly, relatively, slightly, probably, especially, usually, perhaps | Hedging adverbs |
| 244 | relatively, largely, generally, heavily, previously, too, really, already, increasingly, widely | Degree adverbs |

*Adjectives*

| Feat. | Top 10 tokens | Concept |
| --- | --- | --- |
| 14 | major, important, famous, prominent, professional, possible, successful, special, powerful, short | Importance adjectives |
| 45 | separate, direct, subsequent, surrounding, supporting, following, previous, early, earlier, increasing | Sequence adjectives |
| 52 | last, fourth, sixth, fifth, third, earliest, final, next, first, seventh | Ordinal / sequence words |
| 76 | little, good, serious, better, strong, unknown, difficult, significant, certain, close | Evaluative adjectives |
| 138 | soviet, european, roman, royal, british, portuguese, jewish, indian, domestic, italian | Nationality / empire adj. |

*Table 8.* A list of all "pure" features according to Claude for the first seed for the coherent embedding.

| Feat. | Top 10 tokens | Concept |
|---|---|---|
| *Nouns* | | |
| 22 | states, places, sets, matches, times, hours, weeks, plays, months, terms | Plural count nouns |
| 41 | staff, police, workers, administration, soldiers, officers, crew, troops, writers, authorities | Groups of personnel |
| 58 | studies, members, elements, images, sources, characters, levels, stars, pieces, systems | Plural abstract nouns |
| 99 | systems, artists, areas, countries, parts, images, lines, regions, elements, stars | Plural category nouns |
| 135 | drama, fiction, opera, comedy, plot, script, novel, movie, book, story | Narrative genres |
| 151 | school, museum, hotel, academy, theatre, club, institute, station, assembly, battery | Institutional buildings |
| 156 | actor, writer, director, producer, author, coach, singer, critic, manager, commander | Professions |
| 162 | richard, robert, michael, david, thomas, peter, james, edward, paul, john | Male first names |
| 200 | husband, brother, sister, marriage, daughter, birth, son, mother, wife, family | Family / kinship |
| 208 | effects, damage, winds, casualties, rainfall, plans, reports, conditions, orders, evidence | Effects / damage nouns |
| 234 | committee, commission, company, foundation, council, post, party, regiment, battalion, program | Organizational bodies |
| *Places / geography* | | |
| 25 | scotland, ireland, japan, china, manchester, minnesota, australia, paris, canada, chicago | Places (countries / cities) |
| 85 | israel, wales, california, washington, pennsylvania, chicago, croatia, canada, york, texas | Places (countries / states) |
| 254 | london, australia, britain, germany, japan, france, philadelphia, carolina, africa, england | Countries / places |
| *Function words & units* | | |
| 68 | %, inches, metres, feet, mi, °, km, percent, cm, hundred | Units of measurement |
| 69 | when, around, throughout, within, upon, before, during, towards, alongside, via | Temporal / spatial preps |
| 169 | upon, before, towards, within, toward, alongside, through, onto, around, via | Directional prepositions |

# C. Preliminaries

We briefly recap the essential definitions and two practical lemmas that we later use to prove the interleaving.

**Definition C.1.** A *simplicial complex* $K$ is a finite set of non-empty finite sets, that is closed under taking non-empty subsets. The *vertex set* of $K$, denoted $V(K)$, is the set of singletons of $K$. We say $f : K \to L$ is a map between simplicial complexes if it is a map on the vertex set $f : V(K) \to V(L)$ such that it extends to simplices in the following way: if $\sigma \in K$, then $f(\sigma) \in L$.

**Definition C.2.** Let $I$ be a totally ordered set. A *filtered simplicial complex*, $\mathcal{F} := \{F_s\}_{s \in I}$, is a sequence of simplicial complexes such that if $s \leq t \in I$, then $F_s \subseteq F_t$ and $\bigcup_{s \in I} V(F_s)$ is finite. We call $V(\mathcal{F}) := \bigcup_{s \in I} V(F_s)$ for the *vertex set* of $\mathcal{F}$.

**Definition C.3.** Let $(X, d_X)$ be a metric space and $P \subset X$ be a finite set of samples, then the *Vietoris-Rips filtration* of $P$ is the filtered simplicial complex over $\mathbb{R}$, defined by

$$VR_t(P) := \{\sigma \subset P \mid \sigma \neq \emptyset, d_X(x, y) \leq 2t \quad \forall x, y \in \sigma\}$$

at filtration value $t \in \mathbb{R}$.

**Definition C.4.** Let $f, g : X \to Y$ be continuous maps. A *homotopy* between $f$ and $g$ is a continuous map $H : X \times I \to Y$, where $I$ is the unit interval, such that $H(x, 0) = f(x)$ and $H(x, 1) = g(x)$.

**Definition C.5.** Let $\{p_0, p_1, \ldots, p_n\} \subset \mathbb{R}^n$ be a finite set of samples and let $t > 0$ be a scalar. The *geometric realization* of $VR_\epsilon(P)$, denoted $|VR_\epsilon(P)|$, is a topological space constructed as follows:

$$|VR_t(P)| = \bigcup_{\sigma \in VR_t(P)} \left\{ \sum_{p_i \in \sigma} \lambda_i p_i \mid \lambda_i \geq 0, \sum_{p_i \in \sigma} \lambda_i = 1 \right\} \subset \mathbb{R}^n.$$

**Definition C.6.** Let $\mathcal{K} = \{K_t\}_{t \in \mathbb{R}}$ and $\mathcal{L} = \{L_t\}_{t \in \mathbb{R}}$ be filtered simplicial complexes. Let $\delta \geq 0$ be a scalar. We say $\mathcal{K}$ and $\mathcal{L}$ are $\delta$-interleaved if there exist families of maps $\{f_t : K_t \to L_{t+\delta}\}_{t \in \mathbb{R}}$ and $\{g_t : L_t \to K_{t+\delta}\}_{t \in \mathbb{R}}$, such that for any $t \leq s \in \mathbb{R}$, the following diagrams commute up to homotopy, when passing to the realization.

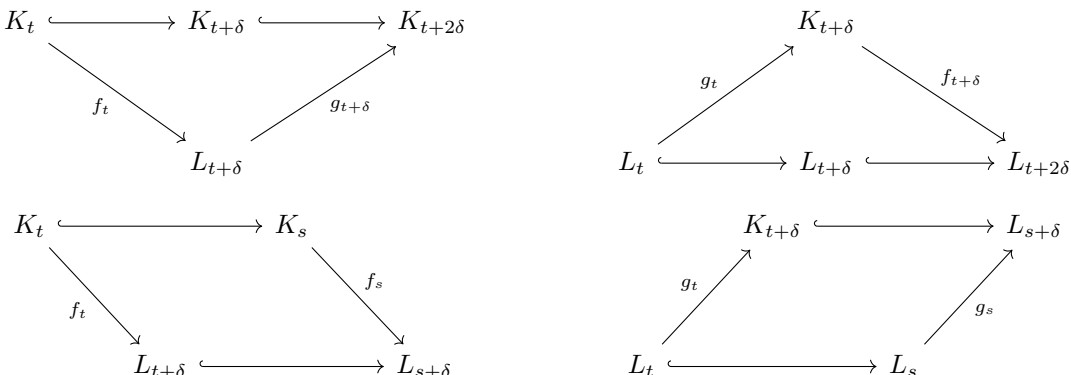

**Definition C.7.** Two simplicial maps $f, g \colon S \to K$ are called *contiguous* if for every simplex $\sigma \subseteq S$ we have that $f(\sigma) \cup g(\sigma)$ is a simplex in $K$.

**Lemma C.8.** *(Spanier, 1994)[Lemma 2, p.130] Two contiguous simplicial maps become homotopic after geometric realization.*

**Lemma C.9.** *Let $K \subset (X, d_X)$ and $L \subset (Y, d_Y)$ be finite subsets of metric spaces $X$ and $Y$. If there is a scalar $\delta \geq 0$ and there are maps $f : K \to L$ and $g : L \to K$ satisfying the following:*

1. *If $d_X(k_i, k_j) \leq 2t$, then $d_Y(f(k_i), f(k_j)) \leq 2t + 2\delta$,*

2. *if $d_Y(l_i, l_j) \leq 2t$, then $d_X(g(l_i), g(l_j)) \leq 2t + 2\delta$,*

3. *if $d_X(k_i, k_j) \leq 2t$, then $d_X(k_i, g \circ f(k_j)) \leq 2t + 4\delta$*

4. *if $d_Y(l_i, l_j) \leq 2t$, then $d_Y(l_i, f \circ g(l_j)) \leq 2t + 4\delta$,*

*then $VR(K, d_X)$ and $VR(L, d_Y)$ are $\delta$-interleaved*

*Proof.* This follows from Lemma C.8. $\square$

## D. Proofs

*Proof of Proposition 3.9.* We only show the first of the statements. By locality we have that

$$Var_{\mathcal{R}}(r_i) = \sum_j w_j^{(i)} \|\phi(r_i) - c_j\|_2^2 \leq \epsilon.$$

Note that

$$\|\phi(r_i) - \Phi(r_i)\|_2^2 = \sum_j w_j^{(i)} \|\phi(r_i) - \Phi(r_i)\|_2^2 \leq \sum_j w_j^{(i)} \|\phi(r_i) - c_j\|_2^2 \leq \epsilon.$$

Here we have used that $\sum_j w_j^{(i)} = 1$ and that $\|\phi(r_i) - \Phi(r_i)\| \leq \|\phi(r_i) - c_j\|$ for all $j$ by definition of $\Phi$. We conclude that $\|\phi(r_i) - \Phi(r_i)\|_2 \leq \epsilon^{1/2}$ by taking the square-root on both sides. $\square$

*Proof of Proposition 3.10.* Let $r_i$ be a row. By definition of $\psi$, being linearly extended by the convex hull we have that

$$\psi \circ \phi(r_i) = \psi(\sum_j w_j^{(i)} c_j) = \sum_j w_j^{(i)} \psi(c_j).$$

Hence,

$$\|r_i - \psi \circ \phi(r_i)\|_2^2 = \|r_i - \sum_j w_j^{(i)} \psi(c_j)\|_2^2 \leq \sum_j w_j^{(i)} \|r_i - \psi(c_j)\|_2^2 \leq \epsilon$$

by using the Jensen's inequality and that $M$ is $\epsilon$-covered. By taking the square root on both sides we arrive at the statement. $\square$

*Proof of Theorem 3.12.* To prove an interleaving we use Lemma C.9, we only prove statement 1 and 3 there as the other two are dual. For 1 we assume that $\|r_i - r_j\|_2 \leq 2t$, then from the 1-Lipschitz assumption on the barycentric maps, we have that $\|\phi(r_i) - \phi(r_j)\|_2 \leq 2t$. By Proposition 3.9 and the triangle inequality we have that

$$\|\Phi(r_i) - \Phi(r_j)\|_2 \leq \|\Phi(r_i) - \phi(r_i)\|_2 + \|\phi(r_i) - \phi(r_j)\|_2 + \|\Phi(r_j) - \phi(r_j)\|_2$$
$$\leq 2t + 2\epsilon^{1/2}.$$

For part 3 in Lemma C.9 we assume that $\|r_i - r_j\|_2 \leq 2t$, then by the triangle inequality we have that

$$\|r_i - \Psi \circ \Phi(r_j)\|_2 \leq \|r_i - r_j\|_2 + \|r_j - \Psi \circ \Phi(r_j)\|_2.$$

By the triangle inequality applied to the last summand and Proposition 3.9 we have that

$$\|r_j - \Psi \circ \Phi(r_j)\|_2 \leq \|r_j - \psi \circ \Phi(r_j)\|_2 + \|\psi \circ \Phi(r_j) - \Psi \circ \Phi(r_j)\|_2 \leq \|r_j - \psi \circ \Phi(r_j)\|_2 + \epsilon^{1/2}.$$

By the triangle inequality on the first summand and Proposition 3.10 we have that

$$\|r_j - \psi \circ \Phi(r_j)\|_2 \leq \|r_j - \psi \circ \phi(r_j)\|_2 + \|\psi \circ \phi(r_j) - \psi \circ \Phi(r_j)\|_2 \leq \epsilon^{1/2} + \|\psi \circ \phi(r_j) - \psi \circ \Phi(r_j)\|_2.$$

By applying the 1-Lipschitz assumption of $\psi$ and Proposition 3.9 on the second summand we have that

$$\|\psi \circ \phi(r_j) - \psi \circ \Phi(r_j)\|_2 \leq \|\phi(r_j) - \Phi(r_j)\|_2 \leq \epsilon^{1/2}.$$

By putting everything together we have that

$$\|r_i - \Psi \circ \Phi(r_j)\|_2 \leq 2t + 3\epsilon^{1/2} \leq 2t + 4\epsilon^{1/2}$$

proving the statement. $\square$

# E. Derived Lipschitz Constants

We give a quick result that allows to express the Lipschitz constant of the barycentric maps in terms of those of the normalization kernel.

**Proposition E.1** (Derived Lipschitz constants). *Let $M \in \mathbb{R}_+^{m \times n}$ with rows $\mathcal{R} = \{r_1, \ldots, r_m\} \subset \mathbb{R}^n$ and columns $\mathcal{C} = \{c_1, \ldots, c_n\} \subset \mathbb{R}^m$. Suppose the normalization kernels $\sigma_R$ and $\sigma_C$, viewed as maps $(\mathbb{R}_+^n \setminus \{0\}, \|\cdot\|_2) \to (\Delta^{n-1}, \|\cdot\|_1)$ and $(\mathbb{R}_+^m \setminus \{0\}, \|\cdot\|_2) \to (\Delta^{m-1}, \|\cdot\|_1)$, have Lipschitz constants $L_{\sigma_R}$ and $L_{\sigma_C}$ respectively. Then the barycentric maps satisfy*

$$\|\phi(r_i) - \phi(r_j)\|_2 \leq K_\phi \cdot \|r_i - r_j\|_2, \qquad \|\psi(c_j) - \psi(c_k)\|_2 \leq K_\psi \cdot \|c_j - c_k\|_2,$$

*where*

$$K_\phi = L_{\sigma_R} \cdot \max_j \|c_j\|_2, \qquad K_\psi = L_{\sigma_C} \cdot \max_i \|r_i\|_2.$$

*Proof.* We prove the bound for $\phi$; the bound for $\psi$ is identical with rows and columns exchanged. By the closed form $\phi(r_i) = \sum_j w_j^{(i)} c_j$ where $w^{(i)} = \sigma_R(r_i)$:

$$\phi(r_i) - \phi(r_j) = \sum_k (w_k^{(i)} - w_k^{(j)}) \, c_k.$$

By the triangle inequality:

$$\|\phi(r_i) - \phi(r_j)\|_2 = \left\| \sum_k (w_k^{(i)} - w_k^{(j)}) \, c_k \right\|_2 \leq \sum_k |w_k^{(i)} - w_k^{(j)}| \cdot \|c_k\|_2 \leq \|w^{(i)} - w^{(j)}\|_1 \cdot \max_k \|c_k\|_2.$$

By the Lipschitz property of $\sigma_R$:

$$\|w^{(i)} - w^{(j)}\|_1 = \|\sigma_R(r_i) - \sigma_R(r_j)\|_1 \leq L_{\sigma_R} \|r_i - r_j\|_2. \qquad \square$$

# F. Relationship to Topological Data Analysis

We would like to discuss the relationship of this paper with ideas and methods developed in the field of topological data analysis. An obvious connection is through the way we formalized our main result in terms of Vietoris–Rips filtrations and interleavings. These concepts are already reviewed in Appendix C. A second deep connection is with the substantial work on relational constructs and in particular Dowker duality and Dowker filtrations. We discuss this connection here.

Given a relation $A \subseteq I \times J$ on the sets $I$ and $J$, the *Dowker row and column complexes* are defined as

$$D_R = \{\sigma \subseteq I \mid \exists j \in J \colon \sigma \times \{j\} \subseteq A\},$$
$$D_C = \{\tau \subseteq J \mid \exists i \in I \colon \{i\} \times \tau \subseteq A\}.$$

The terminology comes from interpreting the relation as a binary matrix where the rows are the elements of $I$ and the columns the elements of $J$. This is also how we can start to see a bridge to the matrix $M$ studied in this paper. The above complexes were first introduced by Dowker in a seminal paper (Dowker, 1952). The main result of that paper establishes a *homology equivalence* between the row and column complexes. This result and its stronger formulation as a homotopy equivalence are today known as *Dowker duality*.

There is a remarkable variety of proofs for this duality result. Without claiming to be exhaustive: Björner (Björner, 1995) gives a proof that relies on a good covering of the row complex in terms of columns and then invokes the nerve lemma; Brun and Salbu (Brun & Salbu, 2023) construct a *rectangle complex* that admits projections to the row and column complexes and then apply Quillen's Theorem A to exhibit these as homotopy equivalences; in another work, Brun and Grinberg (Brun & Grinberg, 2024) use discrete Morse theory to prove the same result; finally, Yoon (Yoon, 2024) gives three different proofs using a Galois connection derived from the relation, a relational join, and a relational product.

What ties all these proofs together is that they rely on some form of fiber lemma and, in order to apply it, some contractibility property. To give an example, in Björner's proof it is crucial that the subcomplexes

$$U_i = \{\tau \subseteq J \mid \{i\} \times \tau \subseteq A\} \subseteq D_C$$

for $i \in I$ form a good cover of the column complex $D_C$.

Another extensive area of research in applied and computational topology concerns possible extensions of Dowker complexes and in particular filtrations of Dowker complexes. One example starts from a non-binary matrix $M$ and then studies the row and column complexes constructed after binarizing $M$ at different threshold values. This leads to filtrations of simplicial complexes and it is possible to lift Dowker duality to an appropriate functorial result in this setting (Chowdhury & Mémoli, 2018; Virk, 2021). Another extension of the classical setting of Dowker duality is introduced by Robinson (Robinson, 2022). There, row and column complexes are combined into cosheaves where one complex is the base space and the other one provides the fibers. The same paper also discusses the notion of *Dowker total weight filtrations*, which annotate the simplices in a Dowker complex with the cardinality of their witnesses and derive filtrations from these.

In the classical Dowker duality setup the involved complexes are agnostic to the exact cardinality of witnesses. This is actually crucial for all proofs of Dowker duality to go through, as it guarantees the aforementioned contractibility conditions that are needed in order to apply the various versions of fiber lemmas. Once we condition the existence of simplices in the complexes on the number of witnesses—like, for example, in the Dowker total weight filtrations—duality breaks down in general.

We believe that here lies an important link to our current work. The relationship of the Vietoris–Rips filtrations of the rows and columns of a matrix $M$ can be understood as a *geometric relaxation* of the classical Dowker duality setting. Similar to the total weight filtrations discussed above, a duality between the row and column pictures of the data may not exist in general in this relaxation. It is then interesting to study conditions on the matrix that recover it, and our coherence regularizer is an attempt at precisely this.

In general, we think that it can be very fruitful to examine all possible proofs of Dowker duality and their detailed intricacies in order to learn about the different failure modes that can arise in geometric relaxations. These failure modes can in turn inspire conditions that prevent such failure and thus guarantee some form of duality—like, for example, an interleaving.

In particular, the different perspectives offered by Yoon (Yoon, 2024) seem very interesting. The relational join is perhaps close in spirit to a blown-up metric space that contains both the rows of a matrix and their barycenters of columns (or vice versa). These could sometimes be interleaved, for example as suggested by the Dowker interleaving result of Chazal et al. (Chazal et al., 2014). Another interesting connection might be Application 4.2 in (Yoon, 2024). In general, spectral sequence computations with (not necessarily good) covers seem very fruitful in studying settings where Dowker duality holds only under appropriate conditions.

A deep connection, in our opinion, is through the general result that $\mathrm{hocolim}\, X_{\mathcal{U}} \simeq X$ for every cover of a topological space $X$, not just good covers. Here $X_{\mathcal{U}}$ is an appropriate simplicial diagram of topological spaces; this is due to Segal (Segal, 1968) and later generalized by Dugger and Isaksen (Dugger & Isaksen, 2004). This result can, for example, replace fiber lemmas in proofs of Dowker duality in order to obtain extensions of Dowker duality to diagrams of appropriate complexes, as pursued by Vaupel and Dunn (Vaupel & Dunn, 2023).

