# OpenReview forum: "Learning Coherent Representations: A Topological Approach to Interpretability"
_ICML.cc/2026/Conference — ICML 2026 regular_

### Official Review · Reviewer_dxnX · 2026-03-09

**Soundness:** 3
**Presentation:** 3
**Significance:** 3
**Originality:** 4
**Overall Recommendation:** 5
**Confidence:** 3

**Summary:**

This paper introduces coherence, a geometric property of non-negative feature matrices inspired by biological neural coding, with the goal of improving the interpretability of latent representations in deep neural networks. It aims to enhance the interpretability of latent representations in deep neural networks by enforcing that each sample attends to geometrically clustered features and each feature activates on clustered samples. The authors formalize coherence using barycentric maps together with Fréchet-variance-based notions of locality and covering, and show that an ϵ-coherent matrix, under a 1-Lipschitz assumption on the barycentric maps, induces a bounded interleaving between the Vietoris-Rips filtrations of samples and features. Based on this framework, the authors derive COH, a differentiable regularizer based on Fréchet variance, and evaluate it on a synthetic two-circle dataset and rotated MNIST. The experiments indicate that COH improves geometric alignment between learned features and known latent factors such as connected components and rotation angle, while also producing a feature space whose topology more closely mirrors that of the sample space, thereby improving interpretability relative to a vanilla autoencoder and L1 sparsity.

**Compliance With Llm Reviewing Policy:**

Affirmed.

**Final Justification:**

The rebuttal resolved my main concerns, especially by expanding the empirical scope and clarifying the theory–practice connection. This strengthens the paper enough for me to raise my recommendation from weak accept to accept.

**Key Questions For Authors:**

1. Can you provide theoretical or algorithmic conditions under which the barycentric maps ϕ and ψ are guaranteed to satisfy a 1-Lipschitz (or more generally, K-Lipschitz) condition for the chosen normalization kernel? If not, have you considered adding an explicit Lipschitz regularization term during training, and if so, could you comment on how this might affect both empirical performance and the interleaving guarantee?
2. How sensitive is the COH regularizer to the choice of non-negative activation function, particularly with respect to sparsity and training stability in the early stages of optimization?
3. The current experiments mainly focus on simple circular structure. Can you provide results showing whether the proposed framework still behaves as intended on data with richer topology, such as multiple circles or torus?

**Limitations:**

The authors adequately and candidly address the limitations of their work in Section 6.They explicitly state that the approach seems most suitable for settings with meaningful geometric structure and non-negative latent representations, which narrows its range of use. The discussion also makes clear that the method may admit multiple coherent solutions without a built-in mechanism for selecting among them. Overall, while the limitations are acknowledged, they also suggest that the current framework is better understood as a specialized geometric approach than as a broadly applicable interpretability method.

**Strengths And Weaknesses:**

Soundness:
The paper presents a technically coherent framework that connects formal definitions of locality, covering, and coherence to a nontrivial topological guarantee via Theorem 3.12. The theoretical development is carefully structured, and the appendices provide useful support for the main propositions and the interleaving result. On the practical side, the proposed COH objective is not merely conceptual: engineering choices such as scale normalization, thresholding, and top-k aggregation make the method trainable in the reported settings. The empirical evaluation is also well aligned with the paper’s central claims. In particular, the authors evaluate the method on datasets with known topological structure, including circles and rotations, introduce concrete and human-interpretable metrics such as component score, MRL, and MRL180, and use persistence diagrams to assess the claimed topological mirroring.

Presentation:
The paper is generally well structured. The authors do a good job of presenting technically dense topological data analysis (TDA) concepts in a reasonably accessible narrative. The visual aids, particularly the UMAP embeddings paired with persistence diagrams (e.g., Figures 3 and 4), effectively illustrate the main topological claims. Proof sketches in the main text also help with readability, while the more detailed proofs are appropriately deferred to the appendix.

Significance:
The paper studies an interesting problem at the intersection of interpretability and geometry, and introduces a distinctive perspective by relating latent samples and latent features through topological alignment. The core idea is meaningful, and the fact that the method is formulated as a regularizer for nonnegative autoencoders makes it potentially useful beyond the specific datasets explored.

Originality:
The paper presents a genuinely distinctive perspective by framing interpretability in terms of topological alignment between latent samples and latent features, rather than focusing only on sparsity or on topology preservation in the sample space alone. In particular, the emphasis on the topology of the feature space itself, and not only the latent sample space, is a fresh and interesting contribution.
While the paper proposes an interesting framework, the connection between theory and practice remains incomplete. The main topological guarantee in Theorem 3.12 relies on a 1-Lipschitz assumption on the barycentric maps, but this condition is not explicitly enforced in the Section 4 COH loss and is instead validated empirically afterward. In addition, the empirical evaluation is limited to small, highly structured datasets with known topology, and the baseline set is relatively narrow.I also noticed what appears to be a notation or dimensional inconsistency around Definition 3.1 (lines 165-166).In addition, I think the relationship between the theoretical notion of ϵ-coherence and the practical top-k, thresholded COH objective could be explained more clearly, particularly with respect to how the practical loss relates to the assumptions required by Theorem 3.12.

---

> ### Author Rebuttal · Authors · 2026-03-31
>
> We would like to thank the reviewer especially for the interesting questions about algorithmic details.
>
> ### 1. Experiments with torus and sphere
> All of the reviewers had a similar remark about the limited scope of our experiments. We agree with this critique and performed much more extensive experimentation now. We thank all of the reviewers for encouraging these additions.
> We have included several new experiments, in addition to the small language model experiments.  \\
> We repeated the experiments we performed with circles but now with tori and spheres. This yielded similar results where we learned coherent representations with latents that were confined to local regions on the geometric structure in question. See: https://anonymous.4open.science/r/Learning-Coherent-Representations-A-Topological-Approach-to-Interpretability-3375/
>
> ### 2. General purpose interpretability metric
>
> We agree that it is not obvious what the ideal general-purpose metric for measuring interpretability is. In cases of circular structures and similar it is easier to suggest something but in cases of no obvious underlying geometry (like the language models we experimented with) we require something more universal. In general, high reconstruction ensures fidelity, while high coherence ensures each latent dimension activates on a contiguous manifold region. We believe that this provides an unsupervised, label-free interpretability objective that, unlike for example DCI, MIG or SAP, requires no ground-truth factors of variation. \\
> In the paper, angular metrics appear only because ground-truth circular structure was available in our experiments, enabling external validation, while for the language model it made more sense to look at other aspects of the embedding matrix. We thank the reviewer and would like to include such a discussion in the article.
>
> ### 3. The Lipschitz condition
> This is a very interesting question. We gave a long reply to a similar question of reviewer Qnnu (reply 1.) and would like to refer to that one.
>
> ### 4. Choice of non-negative activation function
> We thank the reviewer for digging into this detail.
> Indeed, the choice does matter. Practice requires a balance between two extremes. It has to be relaxed enough to not kill off features by pushing the values to where the gradient of the activation function decays. At the same time, it should be sharp enough to facilitate an appropriate amount of sparsity. Empirically, we have found softpplus to be a good compromise, typically with a higher $\beta$ value, like $20$.
>
>
> ### 5. Experiments on language models
> To illustrate the applicability of our method to settings beyond simple geometries we applied our COH loss to token embeddings in BERT-like models. Please see answer 1. to reviewer aco7 for more details on this.
>
> ### 6. Typo in Definition 3.1
> There was a flipped $m$ and $n$ superscript in the definitions of the row and column normalization kernels. We fixed this and thank the reviewer for pointing out the mistake.
>
> ### 7. Relationship of coherence and the practical top-k, thresholded COH objective
> We thank the reviewer to make this good point.
> Theoretically, we only care about the worst offender and should just punish the single worst row/column during training. However, this would be extremely slow and we actually found gradients to be better behaved when we punish the top-$k$ for a reasonable $k>1$. We do not find the model to be particularly sensitive to the exact value of $k$ and have left it as a hyper-parameter.

---

> > ### Author Rebuttal · Reviewer_dxnX · 2026-04-02
> >
> > Thank you for the rebuttal. My concerns have been adequately addressed. The rebuttal expands the empirical scope, clarifies the connection between the practical COH objective and the theoretical framework, and addresses the notation issue I raised. Overall, this strengthens the paper and resolves my main concerns.

---

> > > ### Author Response · Authors · 2026-04-04
> > >
> > > We thank the reviewer for the constructive feedback, which helped improve the paper. We are glad the revisions — including the expanded empirical scope and clarified theoretical connections — resolved the reviewer's main concerns. We would gently encourage the reviewer to consider adjusting their score accordingly.

---

### Official Review · Reviewer_URif · 2026-03-09

**Soundness:** 2
**Presentation:** 2
**Significance:** 3
**Originality:** 3
**Overall Recommendation:** 5
**Confidence:** 4

**Summary:**

This paper introduces a notion of coherence for neural representations, the idea is that features should activate on geometrically contiguous regions of the data manifold to improve interpretability. The paper also makes a theoretical contribution linking rows and columns via topology; it shows that matrices satisfying the proposed $\epsilon$-coherence condition induce an interleaving between the Vietoris--Rips filtrations of the sample and feature spaces. The authors implement this idea through a differentiable loss term and demonstrate its behavior on synthetic data and rotated MNIST.

**Compliance With Llm Reviewing Policy:**

Affirmed.

**Final Justification:**

I thank the authors for the discussion and especially for engaging on the mathematical aspects of their work. I had concerns in my original review on the abuse of terminology and also not enough referencing on the applied and computational topology literature, especially since some of the concepts and techniques used by the authors in their submission have been directly studied previously in the topological data analysis community.  Given the authors' commitment to expand on this discussion and extensively expand their citations in this direction, as well as their commitment to be more careful about terminology, this addresses my main concerns and I have therefore increased my score to 5.

**Key Questions For Authors:**

The paper is motivated by Dowker duality and studies a topological relationship between the row and column spaces of a matrix representation. However, there is a substantial recent literature on Dowker filtrations and topological constructions on relations [1]; [1], in particular, gives three proofs of Dowker duality via Galois connections, relational join, and relational product. Since the setting here also starts from a relation between two sets (samples and features encoded as a matrix):
- Can the constructions proposed in this paper be interpreted in terms of Dowker filtrations or weighted/generalized Dowker complexes
- How do the proposed barycentric maps and coherence conditions relate to the relational constructions used in the TDA literature?

[1] I. Yoon "Dowker duality, profunctors, and spectral sequences" arxiv:2408.13136 (2024)

More specifically,
### Relational join/product:
Two of the proofs of Dowker duality in [1] rely on relational join and relational product constructions. Since the matrix in this work represents a relation between samples and features, these constructions appear conceptually close to the framework proposed here.
- Can the authors comment on whether the coherence conditions or barycentric maps introduced in this work correspond to (or can be reformulated using) relational joins/products between sample and feature relations?
- If so, this might provide an alternative theoretical perspective on the interleaving between row and column filtrations.

### Galois connections:
The third proof of Dowker duality in [1] uses a Galois connection between subsets of rows and subsets of columns induced by the relation. Given that the paper constructs barycentric maps between convex hulls of rows and columns, it would be interesting to understand whether these maps relate to a Galois-type correspondence between the two spaces.
- Do the authors see their framework as a geometric relaxation of such a correspondence or is the relationship purely conceptual?

**Limitations:**

### Relation to current literature:
The paper would benefit from a deeper engagement with the existing literature connecting matrix relations and TDA. In particular, several works studying Dowker filtrations and their applications in TDA appear closely related to the perspective adopted here but are not discussed. For example, work by Facundo Mémoli, Samir Chowdhury, and Iris Yoon (just to name a few) on Dowker filtrations and dual complexes studies topological structures arising from relations between two sets and could provide useful context for the row/column duality explored in this paper. Incorporating these references and clarifying how the proposed framework differs from or extends this line of work would strengthen the positioning of the contribution.

- [2] Michael Robinson. “Cosheaf representations of relations and Dowker complexes”. en. In: Journal of Applied and Computational Topology 6.1 (Mar. 2022), pp. 27–63. issn: 2367-1734. doi: 10.1007/s41468-021-00078-y.
- [3] Xiang Liu, Huitao Feng, Jie Wu, and Kelin Xia. “Dowker complex based machine learning (DCML) models for protein-ligand binding affinity prediction”. In: PLoS Computational Biology 18.4 (Apr. 2022), e1009943. issn: 1553-734X. doi: 10.1371/journal.pcbi.1009943.
- [4] Samir Chowdhury and Facundo M´emoli. “A functorial Dowker theorem and persistent homology of asymmetric networks”. en. In: Journal of Applied and Computational Topology 2.1 (Oct. 2018), pp. 115–175. issn: 2367-1734. doi: 10.1007/s41468-018-0020-6.
- [5] Morten Brun and Darij Grinberg. The Dowker theorem via discrete Morse theory. arXiv:2407.15454 [math].
July 2024. doi: 10.48550/arXiv.2407.15454.

Additionally, there exists more literature on circular and spherical coordinates:
- [6] AJ Blumberg, M Carrière, JH Fung, MA Mandell. Subsampling, aligning, and averaging to find circular coordinates in recurrent time series, arXiv preprint arXiv:2412.18515
- [7] Nikolas C Schonsheck and Stefan C Schonsheck. “Spherical coordinates from persistent cohomology”. In: Journal of Applied and Computational Topology 8.1 (2024), pp. 149–173.
- [8] T Paik, J Park. Circular coordinates for density-robust analysis, arXiv preprint arXiv:2301.12742,

### Regularizer vs Optimization Condition
The paper introduces COH as a regularizer enforcing coherence during training. However, the theoretical results are formulated as properties of matrices satisfying the $\epsilon$-local and $\epsilon$-covered conditions (Definition 3.11), which then imply an interleaving between the row and column Vietoris--Rips filtrations. It would be helpful to clarify the precise relationship between the theoretical conditions and the optimization objective. In particular, since the guarantees apply to matrices that satisfy the coherence conditions rather than to the training dynamics themselves, the term "regularizer" may overstate the theoretical connection. Framing COH more explicitly as an optimization heuristic encouraging approximate coherence would probably be more accurate.

**Strengths And Weaknesses:**

Strengths:
- The paper studies the interpretability of learned representations, which is a central aspect in modern machine learning. More specifically, it asks how to ensure that latent features correspond to geometrically meaningful regions of the data manifold rather than arbitrary activations.
- The notion of coherence provides a natural geometric notion of interpretability connecting ideas from neuroscience, topology, and representation learning.
- The COH loss based on Fréchet variance is relatively simple to compute and can be incorporated into standard autoencoder training pipelines

Weaknesses:
- Limited engagement with closely related literature, see further comments below.
- Conceptual relationship to relational constructions remains unclear, see below.
- The term "regularizer" is strong and could be overclaiming the theoretical guarantees, see below.
- Limited experimental scope: It is unclear whether the approach scales to larger models or more realistic representation learning settings.
- While the method appears to produce features aligned with geometric structure in simple settings, the connection between the proposed coherence property and broader notions of interpretability in machine learning should be discussed more carefully to avoid overclaiming.

---

> ### Author Rebuttal · Authors · 2026-03-31
>
> We would like to thank the reviewer especially for the interest in the deeper mathematical background of our work. Hopefully our answer is a reasonable engagement with this interest and leads to further discussion.
>
> ### 1. Relationship with proofs of Dowker duality and Dowker filtrations
> We thank the reviewer for the many interesting pointers. The short answer is: we believe that all these connections are extremely interesting and we actively work on building mathematically rigorous bridges and developing new theory. We felt that much of this is beyond the scope of a paper that wants to present the practical applicability of these ideas for interpretability in machine learning.
> The longer answer is: we are thrilled to discuss the connections!
> First of all, we believe that there is an important conceptual difference between classical Dowker duality and what we try to accomplish. In the classical Dowker duality setup the involved complexes are agnostic to the "multiplicity of witnesses". This is actually crucial for all proofs of Dowker duality to go through (it guarantees the contractibility condition that is needed in order to apply the various versions of fiber lemmas). In particular this is true for the functorial Dowker duality from the "thresholding" filtration of a non-binary relation. Once we condition the existence of simplices in the complexes on the number of witnesses, like for example in the Dowker total weight filtrations, duality breaks down in general. It is then interesting to study conditions on the matrix that recover it and our coherence is an attempt at this.
> In general, we think that it can be very fruitful to examine all possible proofs of Dowker duality and their detailed intricacies in order to learn about the different "failure modes" that can arise in geometric relaxations (or for example the Dowker total weight filtrations). These failure modes can in turn inspire conditions that prevent such failure and thus guarantee some form of duality (like an interleaving).
> In particular the different perspectives offered by [1] seem very interesting. The relational join is maybe close in spirit to a "blown up" metric spaces that contains both the rows of a matrix and their barycenters of columns (or vice versa). These could sometimes be interleaved (see for example the Dowker interleaving result in "Persistence Stability of Geometric Complexes" by Chazal et al.). Another interesting connection might be Application 4.2 in [1]. In general, spectral sequence computations with (not good) covers seem very fruitful in studying settings where Dowker duality just holds under appropriate conditions. A deep connection in our opinion is through the general result that $\mathrm{hocolim X_{\mathcal{U}}}\simeq X $ for every cover of a topological space $X$ (not just good covers). Here $X_{\mathcal{U}}$ is an appropriate simplicial diagram of topological spaces (this is due to Segal and later generalized by Dugger and Isaksen). This result can for example  replace fiber lemmas in proofs of Dowker duality in order to obtain extensions of Dowker duality to diagrams of appropriate complexes (like in "The bifiltration of a relation and extended Dowker duality", arXiv:2310.11529).
>
> ### 2. Galois connections
> This Galois connection is very closely related to the Galois connection discussed in the context of formal concept analysis  (and formal concept analysis is in turn very related to the problems we try to understand here). We agree that this connection is very interesting in particular and we are currently working on formalizing ideas around it. This is much more easily done in the framework of total weight filtrations on Dowker complexes. We hope to present all our findings with the correct scope in future work. In the present paper we decided to focus on how coherence of a matrix implies an interleaving on VR complexes as we wanted to stress the relationship of locality of features with interpretability.
>
> ### 3. Parametrization
> We explicitly did not use circular or spherical coordinates as our goal was a general-purpose method. Thus, we did not want to engineer it to only work with specific topological signatures, like the cited coordinatization techniques do.
>
> ### 4. Regularizer vs Optimization Condition
> We thank the reviewer for pointing out how our wording makes wrong implications. We agree with this view and will change the wording to something more appropriate in the final version of the paper. We think to simply call COH an *objective* and hope this is appropriate and unambiguous but are happy to discuss alternatives.
>
> ### 5. Limited experimental scope
> We agree that our experimental scope was limited. In the meantime we performed numerous experiments with autoencoders on tori and spheres (see reply 1. to reviewer dxnX) and more importantly on language models without an obvious topology (see reply 1. to reviewer aco7).

---

> > ### Author Rebuttal · Reviewer_URif · 2026-04-02
> >
> > I would like to thank the authors for engaging with my questions, especially on the more mathematical aspects. I was interested to read their thoughts on the questions I raised and do agree that they are beyond the scope of the paper. I was also satisfied with the additional experiments that the authors ran, so this concern is fully resolved. This being said, I still believe the references to (applied and computational) topology in the paper are severely lacking. If the authors are prepared to discuss further this literature (even if it means writing and including a new dedicated section in the appendix, perhaps also mentioning some of the discussions mentioned above) and fix the phrasing as I suggested above, I would be willing to increase my score to 5.

---

> > > ### Author Response · Authors · 2026-04-04
> > >
> > > We thank again the reviewer for the interesting discussion.  We are happy to:
> > > - expand the references to (applied and computational) topology
> > > - fix the phrasing in the paper regarding the regularizer terminology
> > > - include a new dedicated section in the appendix, discussing the connection with Dowker complexes and filtrations and in particular some of the points from above. We will also add an appropriate paragraph to the related work section of the main text.
> > >
> > > We have prepared the section for the appendix and uploaded it to the anonymous repository. See: https://anonymous.4open.science/r/Learning-Coherent-Representations-A-Topological-Approach-to-Interpretability-3375/appendix_tda.pdf

---

### Official Review · Reviewer_aco7 · 2026-03-12

**Soundness:** 4
**Presentation:** 4
**Significance:** 3
**Originality:** 3
**Overall Recommendation:** 5
**Confidence:** 4

**Summary:**

This paper introduces coherence, a geometric property for non-negative representation matrices inspired by neural coding in the brain (grid cells, head direction cells). A matrix is epslion-coherent if every row (sample) maps close to some columns (features) under a barycentric map, and vice versa, ensuring that samples attend to geometrically clustered features and features attend to clustered samples. The authors prove that coherent matrices induce a bounded interleaving between the Vietoris-Rips filtrations of the sample and feature spaces, providing a topological guarantee that both spaces share compatible structure. They derive COH, a differentiable regularizer based on Fréchet variance, and validate it on a synthetic two-circles dataset and rotated MNIST digits, showing that COH produces features whose activation patterns align with interpretable data attributes (rotation angle) far more reliably than vanilla autoencoders or L-regularized baselines.

**Compliance With Llm Reviewing Policy:**

Affirmed.

**Final Justification:**

The authors addressed my major concerns in the rebuttal therefore I have increased my score.

**Key Questions For Authors:**

- Can you provide any experiment on data whose intrinsic topology is not circular? Even a simple dataset with tree-like or toroidal structure would help demonstrate generality.
- What is the wall-clock training time overhead of COH relative to vanilla training?

**Limitations:**

- The experimental results are fairly limited and narrow in scope. It raises questions on the generality of the approach.
- The cost and scalabilty is a concern. It may be limited to only modestly sized latent dimensions.

**Strengths And Weaknesses:**

Strengths:
- The paper provides an elegant and well-described formalism. The definition of coherence is clean: epsilon-local plus epsilon-covering, yielding a bidirectional geometric coupling between sample and feature spaces. The progression from Fréchet variance to covering to the interleaving theorem is logical.
- The neuroscience motivation is strong.  The observation that biological neural codes are interpretable precisely because individual neurons have spatially localized receptive fields, and that this locality is what coherence formalizes, is compelling and distinguishes this work from generic regularization approaches.
- The example illustrations are compelling. The persistence diagrams confirming matching H₁ structure in both sample and feature space are a nice validation of the theory.
- The double-digit experiment admirably shows that COH can converge to qualitatively different solutions (separated vs. merged circles) across seeds. The authors do not hide this — they present both solutions, show both are coherent with interpretable features, and demonstrate that combining COH + L¹ can steer toward a specific (disentangled) solution


Weaknesses:
- The experimental scope is overly narrow. All experiments use either a synthetic toy dataset or rotated MNIST digits with known circular structure — precisely the setting where the method's circle-detecting capabilities shine. There are no experiments on natural images, language data, tabular data, or any setting where the ground-truth topology is not a priori known to be circular.
- Scalability is a concern.  This is acknowledged by the authors but never tested.  The latent dimension tested is 256 with 512 hidden units.  This is modestly sized.
- The three proposed metrics are all designed around circular/angular structure. There is no general-purpose interpretability metric

---

> ### Author Rebuttal · Authors · 2026-03-31
>
> ### 1. Experiments on language models
>
> We added experiments with COH applied to the token embeddings of a small BERT model with 16M params (dim=512, depth=4, heads=4). We mask all but the top 3000 words and train models on the small wikitext-2 dataset using a standard masked language modeling objective with 15\% masking. We have placed copies of figures in:\\
> https://anonymous.4open.science/r/Learning-Coherent-Representations-A-Topological-Approach-to-Interpretability-3375/
>
> We compared three versions: the standard model (Vanilla), a model with non-negative activations (Soft+), and one with non-negative activations that also minimizes the COH loss (COH). All models are trained for $150$ epochs and achieve a very similar accuracy around $51\%$. We have not seen evidence that using COH negatively impacts the models performance yet, but we have not done careful tuning as we solely focus on showing the impact on the token embedding here.
>
>
> We compute the mean cosine distance between the top $50$ words each feature is most active on (top\_word\_dists.pdf on github). From this, we see that the features in the COH model strongly prefer to activate on similar features (mean at $0.145$). This was not the case for the Vanilla (mean at $0.954$) and Soft+ (mean at $0.292$) models. Indeed, for these models, the distances between word pairs that are highly active on the same feature are similar to the distances between arbitrary word pairs. This suggests that features in the Vanilla and Soft+ models are in superposition while the features in the COH model act more like topics in that they are active on highly similar words.
>
> These findings are further confirmed by looking at individual words. For the non-negative models, we can identify the feature that has the highest value for a given word. We can then look at the top words of the identified feature and see if those share the same (or any) meaning as the original word.
>
> | Feature | With Coherence | Without (Non-neg only) |
> |---|---|---|
> | 2 | 3, 4, 2, 36, 34 | week, rating, reception, innings, lot |
> | three | six, two, seven, five, four | the, three, 10, six, 29 |
> | french | roman, american, royal, french, british | french, private, ring, australian, european |
> | mother | father, mother, man, son, woman | mother, janeiro, constitution, yue, virginia |
> | week | day, days, week, minutes, hours | week, rating, reception, innings, lot |
>
> Moreover, nearby features (under euclidean geometry or cosine distance) seem to share similar meaning when using Coh:
>
> e.g., these are the most significant words of nearest features, staring from the first one,
>
> | Feature | Top 5 Words |
> |---|---|
> | 1 | 3, 4, 2, 36, 34 |
> | 2 | 48, 33, 49, 46, 32 |
> | 3 | 53, 65, 54, 34, 55 |
> | 4 | 4, 40, 38, 36, hurricane |
> | 5 | 10th, 100, 6th, 5th, 12th |
>
> This suggests that the found geometry is encouraging some interpretable properties in the features.
> We want to state that not all features are as convincing, but we believe with better tuning, larger model and data set we will be able to do so.
>
>
> We thank the reviewers for pushing us to develop this more involved example.
> It was an interesting week.
> We think this small LM showcases well that the Coherence objective can be applied successfully in a more typical setting, one where there is no obvious topology and still it encourages interpretability in both the word and feature dimensions.
>
>
> ### 2. Experiments on tori and spheres
>
> Please see our reply 1 to reviewer dxnX.
>
> ### 3. Complexity, scaling and wall clock time
> As we have stated under the algorithm section, the COH loss has a complexity of $\mathcal{O}(R^2C + C^2R)$ where $R$ and $C$ are the number of rows and columns of the matrix we want to apply the loss on. From the formulas, scaling is certainly a concern. For larger token spaces we propose simply
> subsample rows or columns at random for each optimization step for the matrix we apply the Coh objective to. In our experiments, this has been sufficient to globally lower the Coh loss and computationally efficient. For the BERT experiment, we deal with a matrix of shape $3000 \times  512$ (word tokens $\times$ feature dims). Here the model using Coh objective uses $16.0s$ per epoch and the non-negative model uses $13.8s$ per epoch. Hence, the increase is 2.2 seconds per epoch (from 13.8s to 16.0s), which corresponds to roughly a 15.9\% increase in training time.
>
> For larger token spaces we propose simply
> subsample rows or columns at random for each optimization step for the matrix we apply the Coh objective to. In our experiments, this has been sufficient to globally lower the Coh loss and remain computationally efficient.
>
>
> ### 4. Coherence + reconstruction as a general purpose interpretability metric
>
> Please see our answer 2. to reviewer dxnX.

---

> > ### Author Rebuttal · Reviewer_aco7 · 2026-04-03
> >
> > I thank the authors for the substantial additional experiments during the rebuttal period. The new results meaningfully address my primary concerns, and I am inclined to raise my score.
> >
> > BERT experiment.  The demonstration that COH produces topic-like features in a setting with no a priori known topology directly addresses my concern about generality beyond circular structure. The cosine distance analysis (mean 0.29 vs. 0.44/0.43) and the qualitative feature tables (e.g., "mother" → father, son, woman) are encouraging. I appreciate the authors' candor that not all features are equally convincing .  The 15.9% wall-clock overhead is also quite reasonable.
> >
> > Tori and spheres. These confirm the method generalizes beyond circles, though they remain in the regime of known low-dimensional manifold structure. Useful but expected given the theory.
> >
> > Overall, the new experiments shift the paper from "elegant theory on toy problems" to "theory with initial evidence of broader applicability." I raise my score accordingly.

---

> > > ### Author Response · Authors · 2026-04-04
> > >
> > > We thank the reviewer for the thoughtful engagement with our new experiments, we appreciate the kind words and the increase in score.

---

### Official Review · Reviewer_Qnnu · 2026-03-13

**Soundness:** 3
**Presentation:** 4
**Significance:** 3
**Originality:** 4
**Overall Recommendation:** 5
**Confidence:** 4

**Summary:**

This paper introduces a coherence regularizer for nonnegative latent matrices. The setup treats the latent activation matrix as a bipartite relation between samples and features, and defines locality and covering conditions so that both sides induce geometrically concentrated neighborhoods on the other through barycentric maps. The main theoretical claim is a conditional interleaving result between the sample-space and feature-space Vietoris-Rips filtrations. The practical contribution is the COH regularizer, evaluated on a two-circles toy problem and on rotated-MNIST single- and double-digit settings. The paper is motivated by the locality of representations sometimes observed in neurosciece (e.g. grid or place cells)

**Compliance With Llm Reviewing Policy:**

Affirmed.

**Final Justification:**

The rebuttal addressed all concerns and increased my confidence in my rating. I think the extension to other non-circle spaces helps the story and the extended LM experiment (reviewer aco7) is very interesting -- even results aside, the inclusion of a good-faith attempt at applying to a non-toy setting gives the reader better contextualization of the method.

**Key Questions For Authors:**

1. What cases might we expect the 1-lipschitz assumption to fail
2. What are some limitations for extending beyond circle-like spaces -- did you not explore these in order to keep the paper scope contained, or might we expect real challenges in scaling this.

**Limitations:**

Yes

**Strengths And Weaknesses:**

### Strengths
- The core idea is interesting. The paper’s main novelty is the bidirectional coupling between sample-space and feature-space topology. The connection to dowker’s theorem and the motivation from neural representations such as grid cells fit together well and give the paper a coherent conceptual story. I suspect this could have substantial influence on the interpretability community.
- The experiments are intuitive, and well matched to the motivation of the paper. Two-circles and rotated-MNIST settings make the geometric and topological claims easy to inspect directly
- The paper is well written and very well presented.

### Weaknesses
- I had a hard time following the importance of the 1-lipschitz assumption in Thm 3.12. It is verified empirically, but I was left unsure when this assumption should be expected to fail, and how much of the paper's theoretical story depends on it.
- The empirical scope is narrow, though I think it is appropriate for the paper. Even so, I did not come away with a strong sense of how the method would behave beyond circle-like settings. Two-circles and rotated MNIST are both variations on S^1. I would have liked to see at least one qualitatively different geometry, such as an interval, a tree-like object, a torus or cylinder, (esp since the authors motivate by e.g. grid cells).
- I also wanted a negative control with weak or no meaningful geometry. That would help clarify whether coherence is recovering real structure when it exists.

---

> ### Author Rebuttal · Authors · 2026-03-31
>
> We sincerely thank the reviewer for the positive reception of our paper and the valuable suggestions and concerns, that we can hopefully address in our answers below.
>
> ### 1. The Lipschitz condition
> The dependence of our interleaving result on a Lipschitz condition is a concern raised by all reviewers.
> First of all, we would like to mention that if the barycentric maps are $K$-Lipschitz we still get an interleaving. This is then just worse by a factor of $K$. Using the triangle inequality one can relate the Lipschitz constant of the Barycentric maps back to those of the normalization kernel. We plan to add this simple derivation to the Appendix of the paper. From there, one can derive bounds for specific choices of normalization kernels. We did this using the Jacobian of the squared L1 normalization kernel and the intermediate value theorem. In general it is hard to estimate all the terms in it though, so the question is not settled.
>
> Second, we would like to add that in all our experiments (including new experiments added to the text) we find the Lipschitz constant to end up very close to 1 *without* explicitly optimizing for that. This is why we never saw the need to add an explicit loss.
>
> This being said, we agree with the reviewers that our interleaving result would be more satisfactory without the Lipschitz condition.
> Despite considerable effort we have not yet succeeded in finding a way to prove the result without it. A promising direction is to derive a bound on the Lipschitz constant directly from the coherence of the matrix. This way we could remove it as an additional assumption in the Theorem. We expect this to be possible since the Lipschitz constant became very close to $1$ in all our experiment. So it is likely that coherence is enough to guarantee it. We are currently working on derivations involving the Wasserstein distance. These are unfinished and not ready to be presented in more detail. So we also want to exercise caution and not make any strong claims.
>
> We would like to stress again, that we see this as an important point and will include an expanded discussion in the main text, likely with some elaboration in the appendix, along with the derivation mentioned above.\\
>
> For larger token spaces we propose simply
> subsample rows or columns at random for each optimization step for the matrix we apply the Coh objective to. In our experiments, this has been sufficient to globally lower the Coh loss and remain computationally efficient.
>
> ### 2. Negative control
> We thank the reviewer for this very reasonable request. We did several training runs of an autoencoder with coherence loss on a random matrix. In all of these runs the coherence loss became small. Studying low-dimensional embeddings of the rows and columns of the latent embedding matrix we did not see any false geometric structure being learned. Please refer to "NOISE.pdf" on  https://anonymous.4open.science/r/Learning-Coherent-Representations-A-Topological-Approach-to-Interpretability-3375/ for details.
> In some of our recent attempts to apply COH to larger models we found that two bi-clusters were created that could be interpreted as false geometry. With proper regularization (e.g. L2) we could avoid this undesirable effect.
>
> ### 3. Beyond circle like spaces
> Since submitting, we have performed several additional experiments. Training autoencoders with COH-loss on toroidal- and spherical data gave similar results to the circular cases, with features activating on coherent parts of the torus or sphere. Please also see reply 1. to reviewer dxnX. We also extensively tested using the COH loss in language models. See answer 1 to reviewer aco7 for more details. Also see: https://anonymous.4open.science/r/Learning-Coherent-Representations-A-Topological-Approach-to-Interpretability-3375/

---

> > ### Author Rebuttal · Reviewer_Qnnu · 2026-04-04
> >
> > Thank you for the reply. Thank you for the clarification on the Lipscitz condition. I think the additional experiments are substantial and round out an excellent story. I maintain my positive score.

---

> > > ### Author Response · Authors · 2026-04-04
> > >
> > > We thank the reviewer again and are glad the clarification on the Lipschitz condition was helpful. We appreciate the positive assessment of the additional experiments and are grateful for the continued support.

---

### Decision · Program_Chairs · 2026-04-30

**Decision:**

Accept (regular)

**Comment:**

The paper proposes coherence, a geometrically motivated principle for non-negative representation matrices aimed at improving interpretability in deep learning. The key idea is to enforce that each sample attends to geometrically clustered features and each feature activates on clustered samples. Coherence is formalized via barycentric mappings with locality and covering conditions on both samples and features. The main theoretical contribution is a proof that ε-coherent matrices induce a bounded interleaving between the Vietoris–Rips filtrations of sample and feature spaces, establishing a principled topological link between the two. On the practical side, the authors introduce a differentiable regularizer based on Fréchet variance, which can be incorporated into standard training pipelines. Empirical results demonstrate the effectiveness of the proposed approach.


The paper presents an interesting and original idea, and the overall narrative is both conceptually and technically sound. The main novelty lies in the coupling of the topological structures of samples and features, which constitutes a meaningful contribution to the interpretability of learned representations. The connection to neuroscience provides a compelling motivation that distinguishes this work from more generic regularization approaches. The experimental results are intuitive and convincing. The paper is well written, clearly structured, and accessible, with an elegant and well-articulated formalism. In particular, the definition of coherence is clean, providing a natural geometric notion. The progression from Fréchet variance to covering arguments and ultimately to the interleaving theorem is logical and well developed. Moreover, the proposed regularizer is relatively simple to compute and can be incorporated into training pipelines. Overall, the idea is supported convincingly by both theoretical analysis and empirical validation.


The Lipschitzness assumption is not explicitly enforced, and its practical importance for the method is somewhat unclear. The empirical validation, while appropriate, could be expanded to include a broader range of data structures, particularly cases with weak or no inherent geometric structure. Scalability may also pose a limitation, and experiments with larger models would help assess this aspect more thoroughly. In addition, the discussion of related work could be more comprehensive, as well as the positioning of this work within broader notions of interpretability in machine learning. The authors acknowledge several of these limitations, which is appreciated.


The authors have addressed many of the reviewers’ concerns to a significant extent, providing clarifications and additional experiments. The general consensus is that the paper represents a meaningful contribution to a timely and interesting problem, offering an elegant and principled approach supported by both theoretical and empirical results. For these reasons, I recommend acceptance, while encouraging the authors to incorporate the reviewers’ feedback to further strengthen the paper.